# Counterfactual Residual Data Augmentation for Regression

## Abstract

Data-driven modeling in real-world regression tasks often suffers from limited training samples, high collection costs, and noisy observations. Inspired by the impact of data augmentation in vision and language, we propose a novel *Counterfactual Residual Data Augmentation (CRDA)* technique for tabular regression. Our key insight is that once a regressor has modeled the systematic component of the data, the remaining noise can be viewed as an invariant residual that remains stable under small perturbations of carefully selected features. We exploit this residual invariance to generate new, yet realistic, training samples, effectively expanding the dataset without requiring additional real data. Our method is model-agnostic and readily applicable to various types of regressors. In experiments across datasets from a variety of benchmark repositories, on average, *CRDA* reduces an *MLP Regressor*'s MSE by **22.9%** and an *XGBoost Regressor*'s MSE by **6.4%**. When compared to existing state-of-the-art data generators and augmentation techniques, CRDA consistently outperforms in MSE reduction. By adding principled counterfactual variations to the training data, our method offers a simple and efficient remedy for noise-prone, small-sample regression settings.

## 1 Introduction

Data scarcity and noise are frequently encountered obstacles in regression tasks across domains such as medicine, finance, and manufacturing. Collecting large-scale, high-quality data can be expensive or impractical, and existing data augmentation techniques, while well developed in computer vision and NLP, often do not translate naturally to tabular regression. As a result, many supervised learning models fail to fully capture the underlying behavior of real-world processes when only limited training examples are available.

In this paper, we propose **Counterfactual Residual Data Augmentation (CRDA)**, a simple and flexible method to bolster regression performance under small data constraints. The core idea is straightforward: *(i)* we train a base predictor (e.g., MLP or XGBoost) on a dataset, *(ii)* identify one or more features whose perturbations do not alter the residual distribution significantly, and *(iii)* generate new samples by modifying those features while preserving the original "noise" or residual component. To illustrate, consider a house price prediction task. A model captures systematic value drivers like location and square footage, while the residual captures unobserved factors like a *bidding war* driven by a specific buyer's urgency. Our key insight is that varying a secondary feature, such as garage finish, changes the systematic price but is unlikely to alter the specific buyer's urgency. CRDA exploits this independence to synthesize a valid counterfactual: a house with a different garage finish, an updated systematic price, but the exact same "bidding war" residual.

**Motivation and Benefits.** Our motivation stems from the difficulty of acquiring sufficient labeled data in many practical applications, coupled with the risk of overfitting when sample sizes are small. A major attraction of *CRDA* is its ability to insert new data points *without* assuming domain-specific transformations or heuristics. Instead, it relies on a learned predictor to separate systematic behavior from noise, then conserves the latter across minor interventions of designated features. As a result, the augmented samples remain consistent with the underlying distributional assumptions, improving model fit and reducing variance. Empirically, we observe double-digit percent reductions in test error for small-sample regression tasks, demonstrating the utility of our approach across a variety of dataset types.

Our work makes the following key contributions:

**(1) New Data Augmentation Framework.** We introduce a model-agnostic strategy for augmenting tabular regression data, centered on counterfactual reasoning and residual invariance.

**(2) Residual Invariance Principle.** We formalize how certain features can be perturbed without corrupting the noise structure, providing insights to guide feature selection.

**(3) Empirical Validation.** We evaluate CRDA on synthetic benchmarks and real-world datasets from standard repositories (e.g., UCI, PMLB), illustrating consistent improvements across neural and ensemble models.

## 2 RELATED WORK

**General Data Augmentation.** Data augmentation refers to the strategy of enlarging or diversifying a training set via synthetic transformations. While central to success in computer vision and NLP (Zhang et al., 2017; Yun et al., 2019; Cubuk et al., 2019), these techniques often rely on *label-preserving* symmetries (e.g. image rotations) or domain-specific invariances (e.g. back-translation in text). However, applying these methods to tabular regression remains non-trivial.

**Tabular and Regression-Specific Augmentation.** Classical oversampling approaches include SMOTE (Chawla et al., 2002) and its regression extensions (Branco et al., 2017), which interpolate between samples but do not necessarily preserve higher-order feature interactions or heteroskedastic noise. Recent advances seek to formalize regression augmentation through geometric properties. For example, RegMix (Hwang & Whang, 2021) optimizes Mixup policies to generate samples within high-density regions of the data manifold, aiming to preserve the underlying structure. Conversely, C-Mixup (Yao et al., 2022) addresses the risk of manifold intrusion by restricting mixing to sample pairs with high label similarity. Closely related is Anchor Data Augmentation (ADA) (Schneider et al., 2023), which extends Anchor Regression to augmentation. ADA identifies "anchors" (by clustering) and generates samples using a first-order Taylor approximation, effectively assuming local linearity within clusters. While these methods enforce geometric regularity (linearity or manifold density), they can struggle in highly non-linear or sparse regimes where local linearity assumptions fail. CRDA aims to avoid this by enforcing *statistical* regularity (residual invariance) instead.

**Deep Generative Models.** Deep generative models offer an alternative by learning the joint distribution to sample entirely new rows. Approaches like CTGAN (Xu et al., 2019), TVAE (Xu et al., 2019), and TabDDPM (Kotelnikov et al., 2023) have shown promise in privacy-preserving data synthesis. However, these models typically treat the target variable as just another column, failing to preserve an instance's specific residual noise. This frequently leads to "realistic" looking samples that degrade predictive performance.

**Residual Bootstrapping.** In statistical literature, residuals have been leveraged extensively for *uncertainty quantification* rather than data augmentation. For example, the residual bootstrap (Efron, 1979) and conformal prediction methods (Barber & Candès, 2021) resample or reuse residuals to construct confidence intervals. Our work repurposes this mechanism for augmentation.

**Causal and Counterfactual Data Augmentation** Data augmentation typically ignores the generative process behind the data, risking unrealistic synthetic examples. Causal-based approaches (Kocaoglu et al., 2018; Arjovsky et al., 2020) propose integrating structural assumptions so that augmentations preserve invariant relationships across environments. This has been explored in computer vision through interventions on object attributes (Mahajan et al., 2023), and in language by editing tokens in a `do`-intervention style. Our method's core principle, preserving an instance-specific noise term while perturbing features, draws a direct parallel to work in reinforcement learning. Lu et al. (2020) showed that next-state samples remain identifiable under mild assumptions (monotonicity and independence in the noise term). Their Theorem 1 establishes that, once the observed outcome fixes a particular noise quantile, reusing that noise in a "what-if" scenario yields a valid counterfactual next-state. Similarly, CRDA treats the calculated residual as an exogenous noise variable that is assumed to be independent of the features being perturbed. This allows us to systematically generate counterfactuals in a way that is more theoretically grounded than purely generative or interpolation-based techniques.

## 3 BACKGROUND

In this section, we provide an overview of the key concepts that motivate our proposed approach.

### 3.1 COUNTERFACTUAL REASONING AND STRUCTURAL CAUSAL MODELS

A *structural causal model* (SCM) (Pearl, 2009; Peters et al., 2017) formalizes how observed variables are generated by underlying data-generating processes (DGP). Formally, an SCM is specified by a tuple $(\mathcal{X}, \mathcal{Z}, F, P_{\mathcal{Z}})$, where:

- $\mathcal{X} = \{X_1, \ldots, X_m\}$ is the set of endogenous (observed) variables,
- $\mathcal{Z} = \{Z_1, \ldots, Z_m\}$ is the set of exogenous (noise) variables with distribution $P_{\mathcal{Z}}$,
- $F = \{f_i\}_{i=1}^m$ is a collection of structural equations, each of the form

$$x_i \; \leftarrow \; f_i\big(pa_i, z_i\big)$$

where $\mathrm{Pa}_i \subseteq \mathcal{X} \setminus \{X_i\}$ denotes the parents of $X_i$.

An SCM induces a graph, which encodes causal relationships (i.e. who influences whom), and the exogenous noise terms capture stochasticity.

**Interventions and causal effects.** A central notion in causal inference is that of an *intervention*, written $\mathrm{do}(\cdot)$ (Pearl, 2009). By applying $\mathrm{do}(X = x')$, one replaces the original structural equation $X \leftarrow f_X(\mathrm{Pa}, Z)$ with a constant assignment $X \leftarrow x'$. This operation severs incoming edges to $X$, thus altering the downstream (child) variables but leaving other aspects of the system intact. Interventions enable us to simulate hypothetical scenarios, often crucial for answering "what if" questions in hindsight.

**Counterfactuals.** Counterfactual reasoning goes a step further by inquiring about hypothetical outcomes of the target variable $Y$ *given a specific realization of the noise variable $Z$*. Concretely, one first infers the actual setting of $Z = z$, then imagines how the outcome $Y$ would change under a hypothetical intervention $\mathrm{do}(X = x')$. This process involves: *(i)* **abduction**, where we infer **z** from the observed data; *(ii)* **action**, where we override the structural assignment for $X$; and *(iii)* **prediction**, where we propagate $z$ through the modified system to obtain the counterfactual outcome distribution $P(Y'|X = x', Z = z)$ (Rubin, 1974; Holland, 1986; Peters et al., 2017).

### 3.2 MAIN ASSUMPTIONS AND THEORY

The core theoretical principle underpinning CRDA is the assumption of *residual invariance*. This principle posits that for a well-specified regression model, the residual noise term remains distributionally constant under interventions on a specific subset of features. We formalize this as follows.

**Assumption 1.** *Let the feature vector $X$ be partitioned into two disjoint subsets, $X = (X_P, X_R)$, where $X_P$ are the features we intend to perturb (the* perturbable *coordinates) and $X_R$ are the features we hold fixed (the* remaining *coordinates). Let $g(X) = \mathbb{E}[Y|X]$ be the true conditional expectation function, and let $Z = Y - g(X)$ be the corresponding structural noise term. We introduce the following condition:*

$$\mathbb{P}(Z \mid X_P, X_R) = \mathbb{P}(Z \mid X_R) \tag{1}$$

*Equation 1 says that the noise $Z$ is conditionally independent of the perturbable features $X_P$ given the fixed features $X_R$.*

**Proposition 1.** *Suppose Assumption 1 holds. Then for any $x_R$ in the support of $X_R$ and any $x_P, x'_P$ in the conditional support of $X_P \mid X_R = x_R$, we have*

$$\mathbb{P}\big(Z \mid X_P = x_P, X_R = x_R\big) \; = \; \mathbb{P}\big(Z \mid X_P = x'_P, X_R = x_R\big)$$

*Equivalently, $\mathbb{P}(Z \mid X_P = x_P, X_R) = \mathbb{P}(Z \mid X_R)$ is constant in $x_P$.* [1]

---

[1]The proof for this proposition can be found in Appendix M.

Assumption 1 is effectively equivalent to modeling the data-generating process with an **additive-noise structural causal model (SCM)**. By defining the regressor $g$ as the conditional expectation, we decompose the outcome $Y$ into a systematic component and a residual component:

$$Y = g(X_P, X_R) + Z|_{X_R}$$

Our assumption simply states that in this structural equation, the noise term $Z$ does not depend on the specific values of the features in $X_P$ once we have conditioned on the features in $X_R$.

**Causal Interpretation and Hidden Confounding.** To better understand what underlying DGPs satisfy Assumption 1, we can examine a passable causal structure. This is visualized in Figure 1a. It requires that the features selected for perturbation, $X_P$, are *exogenous* with respect to the residual mechanism $Z$. Note that $Z$ is allowed to depend on the fixed features $X_R$ (e.g., heteroscedasticity or confounding on $X_R$), as long as it remains independent of $X_P$.

However, recent work highlights that unobserved confounding is a primary driver of distribution shift failures in real-world tabular data (Prashant et al., 2024; Reddy et al., 2025). When a latent confounder $U$ exists, its influence on $Y$ that is not explained by $X$ is absorbed into the residual term $Z$. Consequently, $Z$ acts as a noisy proxy for $U$. Figure 1b illustrates the case where $U$ causes both $X_P$ and $Y$, thereby creating a "backdoor path" $X_P \leftarrow U \rightarrow Y$. Here, a statistical dependence arises between $X_P$ and $Z$, violating Assumption 1. Therefore, the validity of counterfactual augmentation depends on identifying and excluding such confounded features from the perturbation set.

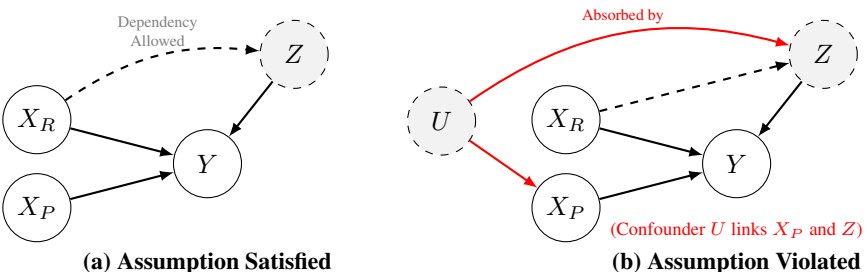

(a) **Assumption Satisfied**                    (b) **Assumption Violated**

Figure 1: **Causal Visualization of Residual Invariance. (a)** The structure satisfies Assumption 1. **(b)** A violation due to unobserved confounding. Here, a latent variable $U$ causes both $X_P$ and $Y$. Since the residual $Z$ absorbs the variation of $U$, a dependency is created between $X_P$ and $Z$, invalidating the augmentation.

## 4 METHOD

In this section, we detail our proposed *Counterfactual Residual Data Augmentation (CRDA)* procedure for tabular regression. The main goal is to augment a limited dataset by synthesizing new samples that remain true to the original noise distribution. Algorithm 1 outlines the full workflow.

### 4.1 ALGORITHMIC OVERVIEW

**Step 1: Data Splitting and Baseline Training.** We begin by splitting the original dataset $\mathcal{D}$ into training and test sets. Let $\mathcal{D}_{\text{train}} = \{(\mathbf{x}_i, y_i)\}_{i=1}^N$ and $\mathcal{D}_{\text{test}} = \{(\mathbf{x}_j, y_j)\}_{j=N+1}^n$. We then fit a *baseline* regressor $g(\cdot)$ on $\mathcal{D}_{\text{train}}$. In practice, this model can be chosen from a variety of families (e.g. MLP, XGBoost) depending on the user's preference.

**Step 2: Residual Computation.** For each training sample $(\mathbf{x}_i, y_i)$, we compute the *residual*

$$z_i = y_i - g(\mathbf{x}_i).$$

Intuitively, $z_i$ captures the latent factors not explained by $g$.

**Step 3: Feature Selection.** To ensure that perturbing a given feature does not spuriously alter the residual structure, we identify and partition the features into $(X_P, X_R)$ so that $X_P$ are eligible for perturbation and $X_R$ are held fixed. Concretely, we select $X_P$ by screening for (approximate) conditional independence of the residual given the remainder:

---

**Algorithm 1** Counterfactual Residual Data Augmentation (CRDA)

---

**Require:** Dataset $\mathcal{D}$ of size $n$,   baseline regressor $g(\cdot)$.
          Hyperparameters: PERTURBATIONRANGE, AUGDATASIZEFACTOR.
1: **Split** $\mathcal{D}$ into $\mathcal{D}_{\text{train}}$ and $\mathcal{D}_{\text{test}}$.
2: **Train** baseline model $g(\cdot)$ on $\mathcal{D}_{\text{train}}$.
3: **for** each $(\mathbf{x}_i, y_i)$ in $\mathcal{D}_{\text{train}}$ **do**
4:     $z_i \leftarrow y_i - g(\mathbf{x}_i)$                                      $\triangleright$ Compute residual
5: **Select partition** $(X_P, X_R)$:
    • PC algorithm to remove features directly connected to $Z$.
    • Correlation check to remove features strongly associated with $Z$.
6: **if** $X_P = \varnothing$ **then**
7:     **return** Baseline $g$                            $\triangleright$ No augmentation possible
8: $\mathcal{D}_{\text{aug}} \leftarrow \varnothing$
9: **for** each $(\mathbf{x}_{i,P}, \mathbf{x}_{i,R}, y_i) \in \mathcal{D}_{\text{train}}$ **do**
10:     **for** $m = 1$ to AUGDATASIZEFACTOR **do**     $\triangleright$ Repeat to generate multiple augmentations
11:         $\mathbf{x}'_{i,P(m)} \leftarrow \text{Perturb}\big(\mathbf{x}_{i,P}; \mathbf{x}_{i,R}, \text{PERTURBATIONRANGE}\big)$
12:         $\mathbf{x}'_{i(m)} \leftarrow \big(\mathbf{x}'_{i,P(m)}, \mathbf{x}_{i,R}\big)$                   $\triangleright$ Hold $X_R$ fixed
13:         $y'_{i(m)} \leftarrow g(\mathbf{x}'_{i(m)}) + z_i$                    $\triangleright$ Preserve residual $z_i$
14:         Add $(\mathbf{x}'_{i(m)}, y'_{i(m)})$ to **augmented set** $\mathcal{D}_{\text{aug}}$
15: **Perform** $K$-fold cross-validation on $\mathcal{D}_{\text{train}}$, comparing *unaugmented* vs. *augmented* models.
16: Collect validation errors $\{e_{\text{unaug}}^{(k)}, e_{\text{aug}}^{(k)}\}_{k=1}^K$.
17: $p$-value $\leftarrow$ WilcoxonSignedRank$\big(\{e_{\text{unaug}}^{(k)}, e_{\text{aug}}^{(k)}\}\big)$
18: **if** $p$-value $\geq \alpha$ **then**
19:     **return** Baseline $g$                       $\triangleright$ No statistically significant improvement
20: $\mathcal{D}_{\text{train}}^{\text{aug}} \leftarrow \mathcal{D}_{\text{train}} \cup \mathcal{D}_{\text{aug}}$
21: **Retrain** a new regressor $g'$ on $\mathcal{D}_{\text{train}}^{\text{aug}}$
22: **return** Augmented regressor $g'$

---

    1. A **causal graph check** applies the Peter-Clark (PC) algorithm (Spirtes et al., 2000) to remove features that are directly connected to the residual $Z$ in the learned graph structure.

    2. A **correlation check** (using Pearson correlation tests) discards any features strongly correlated with $Z$.

The surviving coordinates form $X_P$; the complement forms $X_R$, satisfying Assumption 1. (If none survive, we skip augmentation and return the baseline $g$.)

**Step 4: Input Perturbation.** We expand our dataset by generating AUGDATASIZEFACTOR counterfactual samples per original training point. For each $(\mathbf{x}_{i,P}, \mathbf{x}_{i,R})$ and each $m = 1, \ldots, M$, sample a perturbed $\mathbf{x}'_{i,P(m)}$ and keep $\mathbf{x}_{i,R}$ unchanged:

$$\mathbf{x}'_{i(m)} = \big(\mathbf{x}'_{i,P(m)}, \mathbf{x}_{i,R}\big), \qquad \mathbf{x}'_{i,P(m)} \leftarrow \text{Perturb}\big(\mathbf{x}_{i,P}; \mathbf{x}_{i,R}, \text{PERTURBATIONRANGE}\big).$$

Here, AUGDATASIZEFACTOR $= M$ controls how many new points we generate per sample and can be tuned to balance computational costs against potential gains in generalization. For each $m$, the $Perturb$ operation uses PERTURBATIONRANGE, which we can denote as $p \in (0, 1)$, to sample a single scalar $\delta \sim \text{Unif}\big[-p, p\big]$. The scalar is then used to compute $x'_{i,P(m)} = x_{i,P}(1 + \delta)$. This translates to essentially scaling each chosen feature by a random factor (e.g. $\pm 10\%$), but more sophisticated or domain-specific transformations can be substituted.

**Step 5: Counterfactual Label Assignment.** For each perturbed input $\mathbf{x}'_{i(m)}$, we assign a *counterfactual* label:

$$y'_{i(m)} = g(\mathbf{x}'_{i(m)}) + z_i$$

Crucially, this preserves the original residual $z_i$, thereby keeping the overall noise structure intact under the perturbation (Proposition 1). The newly generated samples $\big(\mathbf{x}'_i, y'_i\big)$ form an augmented set $\mathcal{D}_{\text{aug}}$.

**Step 6: Validation via Cross-Validation and Wilcoxon Signed-Rank Test.** Before committing to a final retraining, we evaluate whether the augmented samples *significantly* improve generalization. Concretely, we run $K$-fold cross-validation on the *original* training data, comparing two models:

1. **Unaugmented model**: trained on the fold's training portion as is.
2. **Augmented model**: trained on the fold's training portion *plus* its augmented points (generated using the same procedure above).

We collect the validation errors (e.g. MSE) across the $K$ folds for both models and perform a non-parametric *Wilcoxon signed-rank test* (Wilcoxon, 1945) on the paired errors. If the resulting $p$-value is below a chosen significance level (e.g. $\alpha = 0.05$), we conclude that augmentation yields a statistically significant improvement; otherwise, we revert to the baseline $g$.

**Step 7: Dataset Augmentation and Retraining.** If the Wilcoxon test finds a significant improvement, we combine $\mathcal{D}_{\text{train}}$ and $\mathcal{D}_{\text{aug}}$ into a single augmented dataset

$$\mathcal{D}_{\text{train}}^{\text{aug}} = \mathcal{D}_{\text{train}} \cup \mathcal{D}_{\text{aug}},$$

We then retrain a *final* model $g'(\cdot)$ with this expanded dataset to use on the test set.

## 4.2 Discussion of Key Design Choices

**Residual Invariance and Causal Assumptions.** The pivotal assumption in our framework is that for certain features, perturbing them does *not* induce a change in the residual distribution. Crucially, CRDA does not require making any causal assumptions about whether $X \to Y$, $Y \to X$ or the absence of confounders. In any regression setting, the joint distribution $P(X, Y)$ can be factorized as $P(Y|X)P(X)$, and the conditional component can be represented by a structural equation $Y = g(X) + Z$, where $g$ is a deterministic function and $Z$ is a noise term that may depend on $X$. The only (non-causal) assumption that we make is Assumption 1 where $Z$ must be independent of $X_P$ given $X_R$. Our method aims to approximate this decomposition by learning the base predictor $g$ and estimating the residual $z = y - g(x)$.

The practical steps in Algorithm 1, such as the PC algorithm and correlation checks, are *empirical heuristics* designed to identify a feature subset $X_P$ for which Assumption 1 is likely to hold. When we reuse a specific residual $z_i$ to construct a counterfactual label $y' = g(x') + z_i$, we are not assuming the noise value itself is invariant, but rather that we are reusing a valid sample from an *invariant noise distribution* $P(Z)$. This is a practical choice that avoids the need to explicitly model the entire noise distribution.

**Model-Agnostic Nature.** Although our algorithm is illustrated with a neural or tree-based baseline, the same counterfactual logic applies to *any* parametric or nonparametric regressor. The key is to view each training outcome as a sum of a learned systematic component and a noise term. Perturbations occur in the input space, but the residual remains anchored to its original data point.

It's worth clarifying that "model-agnostic" means **CRDA can be *attached* to any regression model**, not that it is *guaranteed* to improve every such model. The method includes built-in safeguards to prevent negative impacts. If the residual-feature independence checks (the PC algorithm and Pearson correlation test) fail to identify any suitable features to perturb, or if the final Wilcoxon signed-rank test concludes that the augmentation does not provide a statistically significant improvement, **CRDA gracefully defaults to the untouched baseline model**. This ensures augmentation is only applied when there is empirical evidence of its benefit.

## 5 Experiments

### 5.1 Experimental Setup and Protocol

**High-Level Goal.** Our primary goal is to investigate whether CRDA can reduce test MSE on tabular regression tasks, particularly under conditions of data scarcity. To this end, we evaluate performance across nine benchmark datasets and a synthetic task with a known ground-truth DGP.

**Comparisons and Baselines.**

1. **No Augmentation (Baseline)**: Train the model on the raw data only.

2. **CRDA Augmentation**: Train the same model class on the union of the raw data and the counterfactual samples generated by CRDA.

3. **Generative Model Augmentation**: The baseline model architecture trained on the union of the original data and synthetic samples produced by state-of-the-art tabular data generators

**Hyperparameter Search** We tune *MLPRegressor* and *XGBoostRegressor* with scikit-learn's `RandomizedSearchCV` (Pedregosa et al., 2011) (3-fold) for 20 trials each, totaling *60 fits* per baseline. *CRDA* also introduces three additional knobs: AUGDATASIZEFACTOR, PERTURBATION-RANGE and MAXNUMFEATURESTOPERTURB. We tune these via *Optuna* (TPE sampler) (Akiba et al., 2019) for up to *30* trials, similarly minimizing validation error.

**Datasets and Preprocessing** We consider nine regression datasets (listed in Table 1) from the University of California, Irvine (UCI) Machine Learning Repository, the Penn Machine Learning Benchmarks (PMLB) collection, and Kaggle. These were chosen to represent a variety of numeric, tabular domains and sample sizes. We drop duplicate rows and NaN values, then apply standardization per feature. For each dataset, we produce five training subsets, ranging from $n/5$ up to $n$. All subsets are split 80–20 into training and test sets.

**Evaluation Metrics and Significance Tests** We report the **MSE** (mean-squared error on the held-out test set) for our settings and their relative change (negative values indicate improvement).

$$\Delta\% = 100\,(\text{MSE}_{\text{CRDA}} - \text{MSE}_{\text{baseline}})/\text{MSE}_{\text{baseline}}$$

Significance is assessed with a Wilcoxon signed-rank test across 10 CV folds (Wilcoxon, 1945).

## 5.2 RESULTS AND ANALYSIS

Our main findings are presented in Table 1 and summarized in Figure 2. We see that CRDA provides consistent and substantial reductions in test MSE across nearly all datasets and training set sizes. As shown in Figure 2, the **MLP Regressor benefits most significantly**, achieving an average MSE reduction of **22.9%** across all nine datasets. This highlights CRDA's ability to provide valuable signal for data-hungry neural models. For instance, on the *Parkinson's Monitoring* and *House Price* datasets, MLP models augmented with CRDA see their error reduced by over 30%. The **XGB Regressor** also shows consistent improvement, with an average MSE reduction of **6.4%**.

Table 1 details the performance at different data scales, averaged across 15 unique seeds. Green cells indicate instances where CRDA's cross-validation performance was found to be statistically significant via the Wilcoxon signed-rank test, prompting the final model to be retrained with augmented data. In a few cases (red cells), the test did not find a significant improvement. This built-in safeguard prevents CRDA from being applied where it might not be beneficial. While less frequent, we do note that this filter can be prone to error, particularly depending on sample size, as discussed in Section 6.

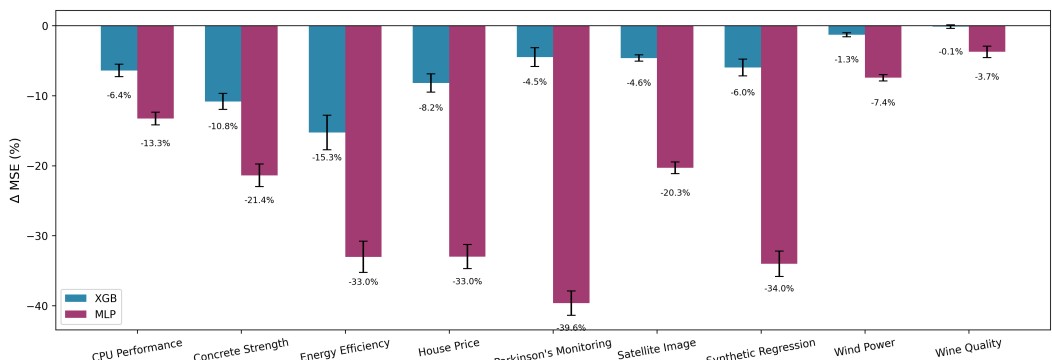

Figure 2: MSE percentage change for each dataset averaged over the five different training-subset sizes reported in Table 1 with error bars corresponding to standard error. Lower is better.

Table 1: Augmentation results for XGB and MLP evaluated and averaged across 15 seeds. Cells are green when data augmentation was more frequently selected to proceed according to the Wilcoxon signed rank test and red otherwise. Lower is better ↓.

| Dataset | Size | XGB ↓ | | | MLP ↓ | | |
|---|---|---|---|---|---|---|---|
| | | $MSE_{baseline}$ | $MSE_{CRDA}$ | $\Delta$ % | $MSE_{baseline}$ | $MSE_{CRDA}$ | $\Delta$ % |
| CPU Performance (Olson et al., 2017a) | 1638 | 0.000968 | 0.000895 | -6.99 | 0.001124 | 0.000869 | -20.24 |
| | 3276 | 0.000885 | 0.000794 | -9.47 | 0.000996 | 0.000853 | -14.03 |
| | 4914 | 0.000772 | 0.000721 | -6.20 | 0.000931 | 0.000822 | -11.31 |
| | 6552 | 0.000726 | 0.000693 | -4.13 | 0.000897 | 0.000795 | -10.48 |
| | 8190 | 0.000738 | 0.000695 | -5.19 | 0.000869 | 0.000780 | -10.23 |
| Satellite Image (Olson et al., 2017b) | 1287 | 0.017775 | 0.016974 | -4.54 | 0.020308 | 0.016285 | -18.36 |
| | 2574 | 0.016361 | 0.015762 | -3.73 | 0.017467 | 0.014547 | -16.69 |
| | 3861 | 0.014599 | 0.013898 | -4.79 | 0.015848 | 0.012115 | -23.14 |
| | 5148 | 0.013661 | 0.013004 | -4.73 | 0.014153 | 0.010760 | -23.72 |
| | 6435 | 0.012540 | 0.011863 | -5.31 | 0.012318 | 0.009888 | -19.66 |
| Wind Power (Haslett & Raftery, 1989) | 1314 | 0.007424 | 0.007208 | -2.82 | 0.007524 | 0.006970 | -7.22 |
| | 2628 | 0.006022 | 0.006034 | 0.20 | 0.006214 | 0.005619 | -9.17 |
| | 3942 | 0.005863 | 0.005785 | -1.33 | 0.005930 | 0.005395 | -9.03 |
| | 5256 | 0.005702 | 0.005624 | -1.40 | 0.005675 | 0.005326 | -6.15 |
| | 6570 | 0.005276 | 0.005218 | -1.08 | 0.005298 | 0.005002 | -5.56 |
| Synthetic Regression (Olson et al., 2017c) | 200 | 0.006520 | 0.005643 | -12.00 | 0.019929 | 0.013874 | -28.80 |
| | 400 | 0.003267 | 0.003124 | -3.16 | 0.006101 | 0.003839 | -36.93 |
| | 600 | 0.002640 | 0.002416 | -7.94 | 0.003213 | 0.002275 | -27.91 |
| | 800 | 0.001646 | 0.001612 | -2.23 | 0.002225 | 0.001402 | -34.12 |
| | 1000 | 0.001524 | 0.001454 | -4.59 | 0.002200 | 0.001231 | -42.33 |
| Concrete Strength (Yeh, 1998) | 201 | 0.007766 | 0.007011 | -8.01 | 0.010334 | 0.007925 | -17.80 |
| | 402 | 0.004929 | 0.004532 | -8.43 | 0.006355 | 0.004956 | -19.83 |
| | 603 | 0.004733 | 0.004269 | -9.75 | 0.006024 | 0.004939 | -17.64 |
| | 804 | 0.003651 | 0.003068 | -15.72 | 0.004966 | 0.003612 | -24.77 |
| | 1005 | 0.002898 | 0.002559 | -12.19 | 0.004224 | 0.003064 | -26.90 |
| Energy Efficiency (Tsanas & Xifara, 2012) | 153 | 0.003991 | 0.003445 | -13.33 | 0.005831 | 0.004258 | -25.10 |
| | 306 | 0.002332 | 0.002061 | -12.20 | 0.003213 | 0.002332 | -28.13 |
| | 459 | 0.001653 | 0.001432 | -10.55 | 0.001884 | 0.001063 | -42.98 |
| | 612 | 0.001281 | 0.000998 | -19.35 | 0.000906 | 0.000522 | -40.71 |
| | 765 | 0.000968 | 0.000761 | -20.96 | 0.000527 | 0.000348 | -28.31 |
| House Price (Community, 2024) | 200 | 0.000785 | 0.000635 | -14.23 | 0.001021 | 0.000572 | -40.57 |
| | 400 | 0.000327 | 0.000310 | -5.39 | 0.000410 | 0.000249 | -37.02 |
| | 600 | 0.000270 | 0.000255 | -4.87 | 0.000290 | 0.000197 | -30.14 |
| | 800 | 0.000243 | 0.000220 | -9.86 | 0.000232 | 0.000158 | -30.32 |
| | 1000 | 0.000196 | 0.000183 | -6.50 | 0.000192 | 0.000138 | -26.97 |
| Parkinson's Monitoring (Tsanas & Little, 2009) | 1175 | 0.000786 | 0.000720 | -8.40 | 0.001650 | 0.001013 | -36.17 |
| | 2350 | 0.000344 | 0.000317 | -6.60 | 0.000799 | 0.000537 | -31.82 |
| | 3525 | 0.000207 | 0.000200 | -2.79 | 0.000484 | 0.000296 | -36.60 |
| | 4700 | 0.000148 | 0.000138 | -6.26 | 0.000422 | 0.000213 | -46.40 |
| | 5875 | 0.000110 | 0.000113 | 1.65 | 0.000259 | 0.000129 | -47.23 |
| Wine Quality (Cortez et al., 2009) | 1063 | 0.020573 | 0.020621 | 0.31 | 0.022912 | 0.022835 | -0.34 |
| | 2126 | 0.014157 | 0.014294 | 1.01 | 0.015394 | 0.014581 | -5.24 |
| | 3189 | 0.013910 | 0.013857 | -0.33 | 0.014784 | 0.014225 | -3.63 |
| | 4252 | 0.013320 | 0.013243 | -0.61 | 0.013862 | 0.013235 | -4.44 |
| | 5315 | 0.013319 | 0.013177 | -1.08 | 0.013972 | 0.013279 | -4.99 |

To better understand how CRDA's effectiveness scales with sample size in a controlled setting, we conducted an experiment on a synthetic DGP with a known ground-truth independence structure:

$$Y = X_1^2 + X_2 X_3 + Z, \qquad \text{where } Z \perp (X_1, X_2, X_3)$$

We generated 50,000 samples and applied CRDA at various sample sizes. The results, shown in Figure 3, reveal that at very low sample sizes ($<2.5k$), CRDA offers minimal benefit because the base predictor is too inaccurate to produce meaningful residuals. Conversely, at very high sample sizes ($>30k$), the baseline model is already so accurate that there is little room for improvement. The *greatest MSE reduction occurs in a "sweet spot"* (between 2.5k and 20k samples), where the baseline model has learned the main signal but still benefits from the localized exploration of the feature space that CRDA provides. This confirms that CRDA is most impactful in low to moderate data-scarce regimes, which are common in real-world applications.

Finally, we benchmarked CRDA against a comprehensive suite of baselines, including regression augmentation methods (C-Mixup (Yao et al., 2022), ADA (Schneider et al., 2023)) and deep generative models (TabDDPM (Kotelnikov et al., 2023), TVAE (Xu et al., 2019), CTGAN (Xu et al., 2019)). As shown in Table 2, CRDA demonstrates superior stability and performance. While geometric methods like ADA and C-Mixup provide gains in specific settings, they exhibit catastrophic

failure modes in others (e.g., increasing MSE by over 100% on *Synthetic Regression* and *Parkinson's*). Similarly, deep generative models significantly degrade performance more often, likely due to difficulties in capturing the precise conditional distribution $P(Y|X)$ required for regression. In contrast, CRDA's residual-preserving mechanism ensures that synthetic samples remain faithful to the underlying noise structure. Across all datasets, CRDA is the only method that reliably improves performance for both XGBoost and MLP models without the risk of significant degradation.

Table 2: The percent MSE change for XGB and MLP base regressors. We compare CRDA against specialized regression augmentations (C-Mixup (Yao et al., 2022), ADA (Schneider et al., 2023)) and generative models (TabDDPM (Kotelnikov et al., 2023), TVAE (Xu et al., 2019), CTGAN (Xu et al., 2019)). Averaged across 10 seeds, reporting standard error. Lower is better ↓.

| Dataset | Model | % MSE Change ↓ | | | | | |
|---|---|---|---|---|---|---|---|
| | | $\Delta_{\text{C-Mixup}}$ | $\Delta_{\text{ADA}}$ | $\Delta_{\text{TabDDPM}}$ | $\Delta_{\text{TVAE}}$ | $\Delta_{\text{CTGAN}}$ | $\Delta_{\text{CRDA}}$ |
| CPU Performance | XGB | $1.7 \pm 1.5$ | $1.9 \pm 0.6$ | $36.5 \pm 4.0$ | $23.6 \pm 3.2$ | $47.5 \pm 3.5$ | **-1.4 ± 0.7** |
| (Olson et al., 2017a) | MLP | $-0.9 \pm 1.0$ | $-0.6 \pm 1.0$ | $27.3 \pm 4.8$ | $30.6 \pm 4.8$ | $141.4 \pm 18.6$ | **-12.0 ± 1.0** |
| Satellite Image | XGB | $6.4 \pm 1.7$ | $1.4 \pm 1.1$ | $10.7 \pm 1.4$ | $13.1 \pm 2.3$ | $8.6 \pm 1.3$ | **-0.7 ± 0.7** |
| (Olson et al., 2017b) | MLP | $-1.0 \pm 2.1$ | $3.3 \pm 2.4$ | $9.9 \pm 3.0$ | $21.8 \pm 4.3$ | $50.5 \pm 5.6$ | **-23.3 ± 1.5** |
| Wind Power | XGB | $-1.9 \pm 0.6$ | $-0.2 \pm 0.3$ | $-0.4 \pm 0.5$ | $4.9 \pm 0.8$ | $8.7 \pm 1.3$ | **-2.6 ± 0.3** |
| (Haslett & Raftery, 1989) | MLP | $4.9 \pm 3.4$ | $14.7 \pm 2.0$ | $-7.1 \pm 1.4$ | $4.9 \pm 1.4$ | $18.7 \pm 2.4$ | **-8.0 ± 1.1** |
| Synthetic Regression | XGB | $141.5 \pm 31.8$ | $18.3 \pm 3.6$ | $25.0 \pm 8.2$ | $117.0 \pm 20.0$ | $158.4 \pm 23.5$ | **2.3 ± 1.8** |
| (Olson et al., 2017c) | MLP | $78.1 \pm 19.2$ | $16.7 \pm 6.4$ | $-19.2 \pm 2.7$ | $68.0 \pm 11.3$ | $191.5 \pm 29.8$ | **-33.3 ± 3.8** |
| Concrete Strength | XGB | **-2.8 ± 1.7** | $-0.1 \pm 1.4$ | $-1.1 \pm 3.0$ | $8.1 \pm 2.7$ | $26.1 \pm 4.4$ | $-1.7 \pm 1.9$ |
| (Yeh, 1998) | MLP | $-4.8 \pm 2.9$ | $-3.8 \pm 1.5$ | $-5.9 \pm 2.9$ | $34.8 \pm 6.3$ | $135.1 \pm 17.7$ | **-15.4 ± 2.3** |
| Energy Efficiency | XGB | $-18.0 \pm 9.1$ | $-20.7 \pm 3.5$ | $3.4 \pm 7.7$ | $-18.3 \pm 8.0$ | **-25.0 ± 6.5** | $-10.7 \pm 3.8$ |
| (Tsanas & Xifara, 2012) | MLP | $11.9 \pm 14.2$ | $-22.9 \pm 4.1$ | $11.1 \pm 11.1$ | $131.8 \pm 32.0$ | $353.7 \pm 52.2$ | **-32.5 ± 5.4** |
| House Price | XGB | $-12.8 \pm 12.6$ | **-42.9 ± 7.4** | $-13.4 \pm 12.9$ | $297.5 \pm 93.6$ | $817.7 \pm 142.1$ | $-12.8 \pm 3.6$ |
| (Community, 2024) | MLP | $-51.0 \pm 5.1$ | **-52.6 ± 5.8** | $-4.3 \pm 16.1$ | $996.5 \pm 286.4$ | $3036.1 \pm 373.9$ | $-42.3 \pm 3.4$ |
| Parkinson's Monitoring | XGB | $105.1 \pm 13.2$ | $89.5 \pm 11.1$ | $286.7 \pm 25.9$ | $434.7 \pm 56.5$ | $596.9 \pm 55.2$ | **-0.3 ± 1.9** |
| (Tsanas & Little, 2009) | MLP | $102.6 \pm 34.9$ | $19.5 \pm 10.1$ | $164.6 \pm 27.1$ | $660.6 \pm 128.0$ | $1280.3 \pm 139.3$ | **-51.1 ± 5.1** |
| Wine Quality | XGB | $-2.0 \pm 0.5$ | $-0.0 \pm 0.6$ | $-2.6 \pm 0.7$ | $-0.5 \pm 0.6$ | $0.6 \pm 0.9$ | **-2.8 ± 0.4** |
| (Cortez et al., 2009) | MLP | $13.1 \pm 5.2$ | $23.6 \pm 2.5$ | **-5.5 ± 0.6** | $-1.2 \pm 0.9$ | $2.2 \pm 0.5$ | $-2.9 \pm 0.8$ |

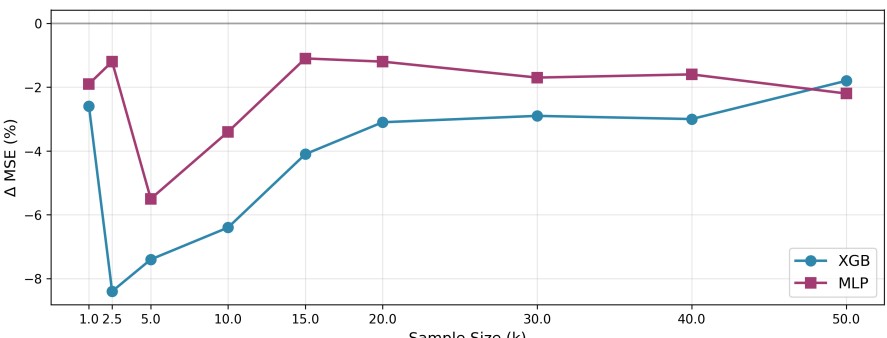

Figure 3: Synthetic sample-size-scaling experiment with a DGP of known independence for the residuals $Z$ and features $X$. For both XGB and MLP base models, we observe a "sweet spot" where CRDA yields the largest MSE reduction (typically at a lower sample size).

## 6 LIMITATIONS

CRDA is currently designed for regression tasks; extending its principles to classification, where residuals are not straightforwardly defined, is a direction for future work.

The core of our method's validity hinges on a key assumption: the model's residual noise, $Z$, is conditionally independent of the features we choose to perturb, $X_P$, given the features we hold fixed, $X_R$ (Assumption 1). In practice, verifying this assumption from finite data is a primary challenge. A poorly fitted base predictor may yield residuals that retain dependencies on all input features, causing interventions on $X$ to break the required noise invariance. To address this, CRDA

employs a two-stage filtering process. First, we use the PC algorithm and a Pearson correlation test as a *risk-control heuristic* to screen for candidate features that are likely to satisfy Assumption 1. We acknowledge this screen is imperfect; the PC algorithm can fail in the presence of unobserved confounders, and correlation tests may not detect non-linear dependencies.

However, these filters are not intended to be infallible but rather a practical first line of defense. The theoretical guarantees of the PC algorithm, for instance, are well-studied; under standard assumptions, its error probability of incorrectly identifying an edge decays exponentially with sample size (Kalisch & Bühlman, 2007). This sample consistency suggests that the risk of our filter admitting a feature that violates Assumption 1 diminishes as more data becomes available. More importantly, CRDA incorporates a second, decisive *safety gate*: the Wilcoxon signed-rank test. This test evaluates the realized impact of augmentation on a validation set. If the generated samples do not yield a statistically significant improvement, the augmentation is discarded. This fail-safe mechanism ensures that we either improve the baseline or abstain from augmentation, thereby mitigating the risk of performance degradation from an imperfect initial screen.

To address concerns that a strong base learner is required, we include *linear regression* experiments in Appendix G. In every dataset/fold, the Wilcoxon gate produced $p > 0.05$, so CRDA *abstained*. If one *ignores* the gate and forces augmentation, performance generally degrades or does not improve, illustrating that the safety checks are beneficial and block harmful augmentation for weak baselines.

Finally, the performance of CRDA is sensitive to both dataset size and the choice of the base predictor. For very large datasets, the need for augmentation diminishes, and CRDA offers little benefit. Conversely, if a dataset is too small, the base model may be too weak to produce meaningful residuals that are even approximately independent of the features, and our statistical tests will lack power. This can be seen in our sample-size-scaling experiment in Figure 3.

## 7 CONCLUSION

We described a new data augmentation technique for regression called CRDA. CRDA is model-agnostic and it does not assume any domain knowledge such as specific transformations that preserve labels. Instead, it leverages counterfactual reasoning and the invariance of the residual noise distribution. We demonstrated the effectiveness of CRDA in data scarce regression tasks where it helped improve predictions made by representative base predictors including XGBoost and multi-layer perceptrons. We also displayed substantially stronger and more reliable results when compared to state-of-the-art tabular data generators.

Several directions for future work remain. The first is to extend CRDA to classification tasks. The key challenge is due to the non-numeric nature of residuals, though embedding-based transformations offer a potential path. The second would be to explore alternative methods in our feature partitioning step, such as formal proximal causal inference techniques. This may enable CRDA to better adjust for hidden factors rather than simply discarding confounded features.

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

# APPENDICES

## A IMPLEMENTATION DETAILS

All experiments were conducted in `Python`, leveraging standard libraries for machine learning and hyperparameter optimization. We used `scikit-learn` for the `MLPRegressor` baseline (Pedregosa et al., 2011), `XGBoost` for the `XGBoostRegressor` baseline (Chen & Guestrin, 2016), and `Optuna` for tuning CRDA's specific hyperparameters (Akiba et al., 2019).

Our experimental protocol uses 10-fold cross-validation (CV), and we seed all random number generators to ensure reproducibility. Computations were performed on a single **AWS c7i.24xlarge** instance equipped with 96 vCPUs and 192 GB of RAM. To facilitate the complete reproduction of our findings, we have made the full study logs, JSON configuration files, and source code available in the project repository, **code included in supplementary material**.

## B DATASET SUMMARY

Table 3 lists the nine tabular-regression benchmarks used in the paper, with their total sample count, dimensionality (# numeric columns), and provenance repository.

Table 3: Basic statistics of the evaluation datasets, including number of samples ($n_{\text{samp}}$) and number of features ($n_{\text{features}}$).

| Dataset | $n_{\text{samp}}$ | $n_{\text{features}}$ | Source |
|---|---|---|---|
| CPU Performance | 8192 | 12 | PMLB (Olson et al., 2017a) |
| Satellite Image | 6435 | 36 | PMLB (Olson et al., 2017b) |
| Wind Power | 6574 | 14 | UCI (Haslett & Raftery, 1989) |
| Synthetic Regression | 1000 | 10 | PMLB (Olson et al., 2017c) |
| Concrete Strength | 1005 | 8 | UCI (Yeh, 1998) |
| Energy Efficiency | 768 | 9 | UCI (Tsanas & Xifara, 2012) |
| House Price | 1000 | 7 | Kaggle (Community, 2024) |
| Parkinson's Monitoring | 5875 | 20 | UCI (Tsanas & Little, 2009) |
| Wine Quality | 5318 | 11 | UCI (Cortez et al., 2009) |

## C COMPLETE EXPERIMENTAL PROTOCOL

**Configuration Objects.** We centralize all experiment settings (e.g. dataset path, model type, global seed, CRDA knobs) in a Python class `Config`. Each run instantiates a `Config` with specific arguments and passes it to our `Experiment` harness, which saves the resulting configuration to a JSON file for reproducibility.

Listing 1: Example truncated config file for an XGB run.

```
{
  "baseline": "xgboost",
  "dataset_path": "../data/WineQuality.csv",
  "sample_sizes": [1063, 2126, 3189, 4252, 5315],
  "ignore_filter": true,
  "hyperparam_tune": true,
  "results_dir": "../experiments/WineQuality",
  "...": "More fields omitted (test_size, num_seeds, p_wilcoxon_threshold
    , etc.)"
}
```

**Key Fields and Usage**

- **Model parameters:** `baseline` can be set to "mlp" or "xgboost"; we do not alter other hyper-parameters (those are tuned via `RandomizedSearchCV`).

- **CRDA knobs:** `aug_data_size_factor`, `max_n_features_to_perturb` and `max_perturb_percent`. These are also tunable parameters (via `Optuna`). They specify how many counterfactual samples to generate, how many perturbable features we perturb and by how much; see Section E.

- **Data splits:** `sample_sizes` enumerates partial subsets of a dataset (e.g. $\frac{n}{5}, \ldots, n$), and `test_size` sets the final train–test ratio.

- **Miscellaneous toggles:** `hyperparam_tune` (whether to run a cross-validated search), `ignore_filter` (bypass CRDA's feature independence checks), `save_plots`, etc.

For each experiment, the `Experiment` class reads the `config` object, runs the pipeline (training, augmentation, evaluation), and dumps logs plus final results in a timestamped directory. By reloading `config.json` via `Config.from_dict`, one can exactly reproduce the same run.

## D   HYPER-PARAMETERS AND SEARCH SPACES

The two baseline families–**MLPRegressor** and **XGBoostRegressor**–share a hybrid strategy: we *fix* well-established architectural or optimisation knobs to textbook defaults, while *searching* over the handful of hyper-parameters that most strongly drive bias–variance trade-offs. This mirrors common practice in tabular ML benchmarks (Fernández-Delgado et al., 2014; Friedman, 2001) and keeps the search budget (20 trials per 3-fold, per dataset, per baseline) focused on the levers that matter.

**Why these choices?**   For MLPs we retain the ReLU–Adam recipe that has been shown to be robust for small/medium tabular tasks (Goodfellow et al., 2016). We enable *adaptive* learning-rate and early stopping to guard against over-training, and explore only depth/width ('hidden_layer_sizes') and three learning-dynamics scalars ($\alpha$, learning_rate_init, tol). For XGBoost we follow the histogram grow policy ("`tree_method=hist`") that is memory-friendly on CPUs, fix 1 000 boosting rounds (with early stopping inside the CRDA loop), and search the usual five knobs that govern tree shape, sampling and shrinkage.

Median wall-clock per trial on a `c7i.24xlarge` is ∼9s (MLP) and ∼4s (XGB).[2] Tables 4 and 5 enumerate the *search priors* together with the *modal* best value across datasets.

Table 4: MLPRegressor hyper-parameters searched with RANDOMIZEDSEARCHCV. Ranges use log-uniform (LogU) or categorical priors.

| Parameter | Prior / Range | Modal best |
|---|---|---|
| hidden_layer_sizes | $\{(128, 64, 32), (128, 64), (64, 32), (64, )\}$ | (128,64,32) |
| $\alpha$ (L2) | $\text{LogU}(10^{-5}, 10^{-3})$ | 0.00040 |
| learning_rate_init | $\text{LogU}(10^{-3}, 10^{-2})$ | 0.00942 |
| tol | $\text{LogU}(10^{-5}, 10^{-4})$ | 0.00009 |
| *Fixed for all runs* | | |
| activation | relu | |
| solver | adam | |
| batch_size | 32 | |
| max_iter | 1 000 | |
| learning_rate | adaptive | |
| early_stopping | true | |
| validation_fraction | 0.10 | |
| n_iter_no_change | 20 | |

---

[2]Full timing logs available in `experiments/full_reproduction.ipynb`.

Table 5: XGBoostRegressor hyper-parameters searched with RANDOMIZEDSEARCHCV. Log-spaces are base-10.

| Parameter | Prior / Range | Modal best |
|---|---|---|
| learning_rate | $\log_{10}[10^{-3}, 10^{-1}]$ (10 pts) | 0.02154 |
| max_depth | $\{3, 4, 6\}$ | 6 |
| min_child_weight | $\{1, 5\}$ | 5 |
| subsample | $\{0.7, 1.0\}$ | 0.7 |
| colsample_bytree | $\{0.7, 1.0\}$ | 0.7 |
| reg_lambda | $\log_{10}[10^{-3}, 10^{1}]$ (6 pts) | 0.03981 |
| *Fixed for all runs* | | |
| objective | reg:squarederror | |
| tree_method | hist | |
| n_estimators | 1 000 | |
| reg_alpha | 0.0 | |
| early_stopping_rounds | 20 | |

## E  CRDA KNOB SELECTION & SENSITIVITY

CRDA exposes three *augmentation knobs*. During a 30-trial OPTUNA–TPE search (per dataset, per baseline) we sample from the priors in Table 6; all other implementation details are inherited from Algorithm 1 (main paper).

- **max_n_features_to_perturb**  controls *how many* invariant features are jointly edited, trading off sample realism against diversity.
- **aug_data_size_factor**  decides the # of counterfactuals per real point; values $< 1$ mitigate class-imbalance–style bias, whereas $> 1$ favors variance reduction.
- **max_perturb_percent**  sets the half-width of the $[-p, +p]$ uniform scaling band; larger $p$ injects broader *counterfactual sweep* but risks violating local linearity assumptions of the residual.

These three parameters explain the vast majority of between-trial variance in validation MSE, so limiting OPTUNA to a small budget remains effective. Median trial time is $\sim 7.5$s (MLP) and $\sim 3.7$s (XGB).

Table 6 reports the *modal* best value across the nine benchmarks.

Table 6: CRDA augmentation knobs: search priors and modal best values.

| Knob | Search prior / range | Modal best |
|---|---|---|
| max_n_features_to_perturb | $\{1, 2, 3, 4, 5\}$ | 2 |
| aug_data_size_factor | $\{0.50, 0.75, 1.00, 1.25, 1.50\}$ | 1.25 |
| max_perturb_percent | $\{0.10, 0.20, \ldots, 1.00\}$ | 0.7 |

ONE-DATASET SWEEP (HOUSE PRICE)

For illustration, we fix two of the three knobs at their modal best values (from Table 6) and systematically vary the remaining knob. Figures 4 and 5 show the resulting percentage change in MSE (*lower* is better) on the *House Price* dataset, averaged over five random seeds. We make several observations:

- **Augmenting data size** (aug_data_size_factor) appears more beneficial for MLP, presumably because additional training samples reduce overfitting; by contrast, XGB sees weaker or even mixed effects here, consistent with the notion that tree ensembles can already leverage smaller sets effectively.

- **Number of perturbed features** (`max_n_features_to_perturb`) shows an opposite preference: XGB yields stronger gains when more features are jointly modified, whereas MLP performance degrades if we perturb too many simultaneously (likely hurting the local consistency of the residual).

- **Perturbation magnitude** (`max_perturb_percent`) also diverges across baselines: larger scales help XGB discover more diverse synthetic points, but MLP tends to prefer smaller shifts in order to maintain stable gradients in training.

In short, although *both* models benefit from CRDA overall, their ideal hyper-parameter configurations differ. This shows the importance of model-aware tuning for effective data augmentation.

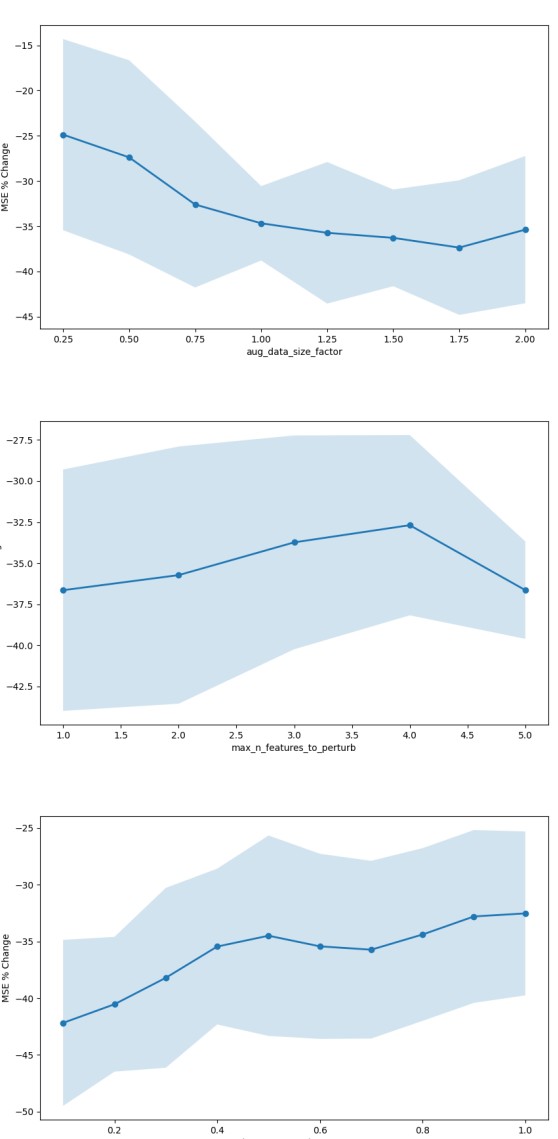

Figure 4: CRDA knob–sensitivity on the **MLP** baseline (HousePrice dataset).

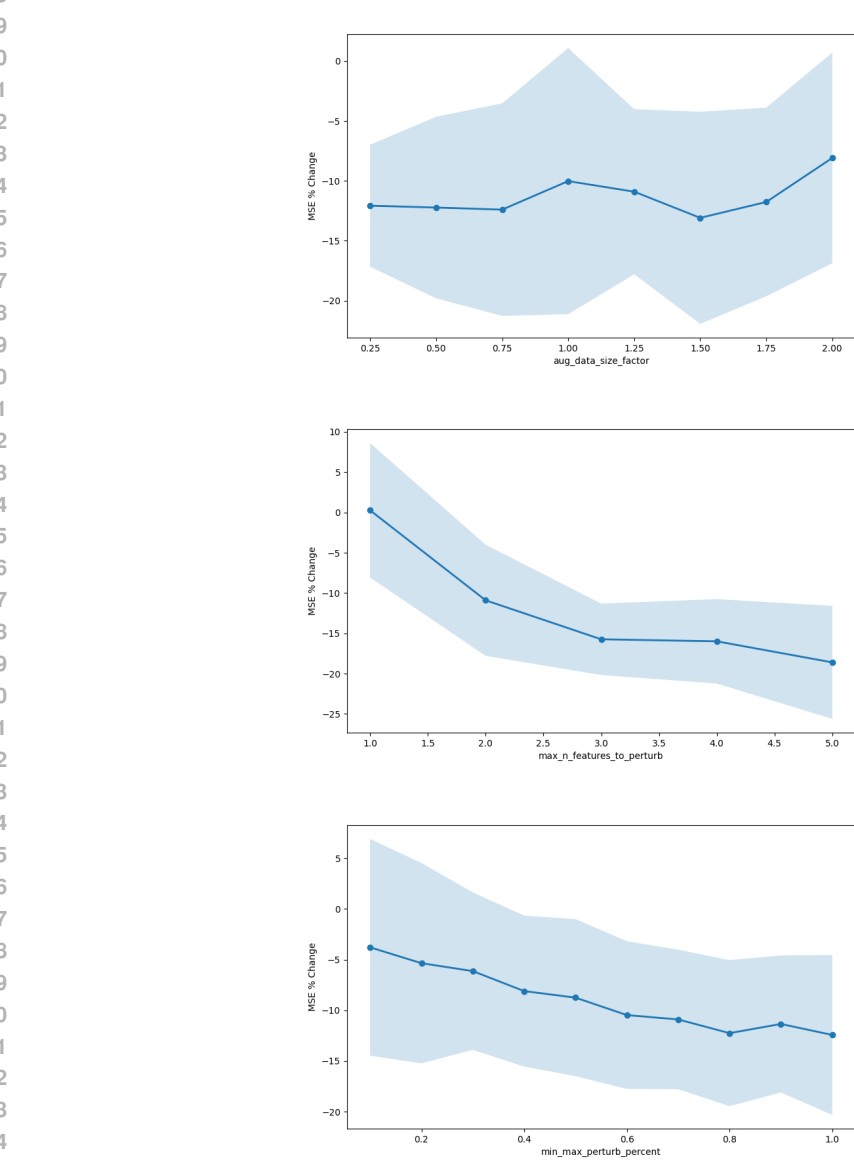

Figure 5: CRDA knob–sensitivity on the **XGB** baseline (HousePrice dataset).

# F VALIDATION OF ASSUMPTIONS AND COMPONENT ANALYSIS

In this section, we provide a deeper analysis of the CRDA framework. First, we empirically validate the core residual independence assumption using Mutual Information. Second, we perform ablation studies to demonstrate the benefits of applying CRDA compared to simplified baselines.

## F.1 EMPIRICAL VALIDATION OF RESIDUAL INDEPENDENCE

A core theoretical assumption of CRDA (Assumption 1) is that the residual noise $Z$ is conditionally independent of the features selected for perturbation ($X_P$), i.e., $P(Z|X_P, X_R) = P(Z|X_R)$. To validate this assumption and assess the effectiveness of our PC-algorithm/Correlation filter, we conducted an analysis measuring the Mutual Information (MI) between the residuals and the features. Mutual Information is an empirical estimator of the KL-Divergence $D_{KL}(P(Z,X)||P(Z)P(X))$; a value of zero indicates perfect independence.

We performed this evaluation across all 9 benchmark datasets using the XGBoost regressor over 15 random seeds. For each run, we calculated the MI (using the Kraskov KSG estimator (Kraskov et al., 2004)) for the set of features *Selected* ($X_P$) by CRDA versus the set of features *Rejected* ($X_R$).

The results are presented in Table 7. We observe that for datasets showing stronger feature-residual dependence (e.g., Energy Efficiency, House Price), the features rejected by our filter display significantly higher Mutual Information with the residuals (up to $\approx 3\times$ higher) than the selected features. This confirms that the filter effectively identifies and removes features that would violate the independence assumption. For other datasets (e.g., Wine Quality, Wind Power), the MI scores for both selected and rejected features are uniformly low, indicating that the residuals are naturally independent of the features in these domains, and the filter correctly permits a wider range of perturbations.

Table 7: Evaluation of Feature-Residual Independence via Mutual Information (MI). We report the MI (in nats) between the model residuals $Z$ and the features $X$, comparing features **Selected** by CRDA vs. those **Rejected**. Results are averaged over 15 seeds with standard errors. The **Ratio** column highlights the effectiveness of the filter in reducing divergence (higher is better).

| Dataset | Selected Features ($X_P$) (Lower is better) | Rejected Features ($X_R$) (Higher implies dependence) | Divergence Ratio ($MI_{Rej}/MI_{Sel}$) |
|---|---|---|---|
| **House Price** | $\mathbf{0.0056 \pm 0.0020}$ | $0.0155 \pm 0.0023$ | $\mathbf{2.75\times}$ |
| **Energy Efficiency** | $\mathbf{0.0054 \pm 0.0012}$ | $0.0160 \pm 0.0023$ | $\mathbf{2.94\times}$ |
| **Parkinson's Monitoring** | $\mathbf{0.0054 \pm 0.0006}$ | $0.0103 \pm 0.0007$ | $\mathbf{1.92\times}$ |
| **Synthetic Regression** | $\mathbf{0.0073 \pm 0.0013}$ | $0.0136 \pm 0.0025$ | $\mathbf{1.85\times}$ |
| Concrete Strength | $0.0203 \pm 0.0024$ | $0.0320 \pm 0.0035$ | $1.58\times$ |
| CPU Performance | $0.0136 \pm 0.0010$ | $0.0144 \pm 0.0014$ | $1.06\times$ |
| Wine Quality | $0.0110 \pm 0.0009$ | $0.0136 \pm 0.0011$ | $1.23\times$ |
| Wind Power | $0.0065 \pm 0.0007$ | $0.0084 \pm 0.0007$ | $1.29\times$ |
| Satellite Image | $0.1031 \pm 0.0013$ | $0.1149 \pm 0.0009$ | $1.11\times$ |

## F.2 ABLATION STUDIES

To verify that the independence assumption verified above translates to performance gains, we compare CRDA against two simplified ablation baselines:

- **Global Perturbation:** All features are perturbed randomly ($X_P = X$), ignoring the PC-algorithm and correlation checks.
- **Label Invariance:** Features are perturbed, but the label is kept fixed ($y' = y$), rather than recalculating $y' = g(x') + z$.

Table 8 presents the percentage change in MSE ($\Delta\%$) relative to the unaugmented base regressor across 3 representative datasets. CRDA consistently yields the largest error reduction. Notably, simple baselines often yield negligible improvements or even degrade performance (positive $\Delta\%$).

Table 8: Ablation results on Synthetic Regression, Energy Efficiency, and Parkinson's Monitoring datasets. Values represent the percentage change in MSE ($\Delta\%$) relative to the unaugmented baseline (lower is better). Results are averaged over 5 seeds with standard errors.

| Dataset | Model | MSE $\Delta\%$ Change ($\downarrow$) | | |
|---|---|---|---|---|
| | | Global Perturbation | Label Invariance | CRDA |
| Synthetic Regression | MLP | $-16.12 \pm 4.30$ | $-12.44 \pm 32.43$ | $\mathbf{-38.94 \pm 4.02}$ |
| | XGB | $+1.21 \pm 2.10$ | $-1.02 \pm 1.33$ | $\mathbf{-3.62 \pm 1.93}$ |
| Energy Efficiency | MLP | $-14.50 \pm 3.86$ | $-2.65 \pm 2.47$ | $\mathbf{-38.84 \pm 5.99}$ |
| | XGB | $-7.15 \pm 9.75$ | $-5.28 \pm 8.78$ | $\mathbf{-17.45 \pm 5.16}$ |
| Parkinson's Monitoring | MLP | $-13.50 \pm 8.90$ | $+0.55 \pm 19.04$ | $\mathbf{-58.40 \pm 5.16}$ |
| | XGB | $+0.36 \pm 1.57$ | $-3.09 \pm 4.35$ | $\mathbf{-7.82 \pm 2.47}$ |

# G LINEAR REGRESSION BASE PREDICTOR STUDY

In order to test our method against weaker base predictors; where separate systematic signal cannot be cleanly separated from noise, possibly violating our assumptions; we selected linear regression. Using the same 15 seeds and data settings as the main experiment, we conducted this study to observe how CRDA behaves.

Table 9 reports the averages and standard errors for baseline MSE, CRDA MSE, their percentage change ($\Delta$ %) as well as the $p - values$ from the Wilcoxon signed-rank test for every dataset and sample size subset.

We see that CRDA's filters *rejected* every single fold. Recall that for the Wilcoxon signed-rank test, if the p-value is above the 0.05 threshold, CRDA stops. We still report the $\Delta$ % if we had ignored the filter and observe how CRDA hurts here. CRDA therefore protects against weaker baselines, further illustrating how model-agnostic does not imply *always helpful*.

Table 9: Augmentation results for Linear Regression. Cells are green when data augmentation was selected to proceed according to the Wilcoxon signed rank test and red otherwise. Lower is better for the $\Delta$ MSE % change ↓.

| Dataset | Size | Linear Regression | | | |
|---|---|---|---|---|---|
| | | $\text{MSE}_{\text{baseline}}$ | $\text{MSE}_{\text{CRDA}}$ | $\Delta$ % ↓ | p-value |
| CPU Performance | 1638 | 0.011094 ± 0.000870 | 0.011066 ± 0.000842 | 0.04 ± 0.58 | 0.461 ± 0.035 |
| | 3276 | 0.010935 ± 0.000576 | 0.011012 ± 0.000604 | 0.58 ± 0.41 | 0.506 ± 0.038 |
| | 4914 | 0.009789 ± 0.000295 | 0.009784 ± 0.000296 | -0.05 ± 0.23 | 0.452 ± 0.038 |
| | 6552 | 0.009834 ± 0.000360 | 0.009887 ± 0.000361 | 0.55 ± 0.18 | 0.515 ± 0.029 |
| | 8190 | 0.009705 ± 0.000347 | 0.009709 ± 0.000346 | 0.05 ± 0.10 | 0.456 ± 0.040 |
| Satellite Image | 1287 | 0.042119 ± 0.000653 | 0.042185 ± 0.000658 | 0.16 ± 0.14 | 0.240 ± 0.037 |
| | 2574 | 0.041148 ± 0.000505 | 0.041131 ± 0.000504 | -0.04 ± 0.07 | 0.264 ± 0.032 |
| | 3861 | 0.040646 ± 0.000388 | 0.040666 ± 0.000393 | 0.05 ± 0.05 | 0.275 ± 0.025 |
| | 5148 | 0.040154 ± 0.000304 | 0.040183 ± 0.000296 | 0.08 ± 0.05 | 0.393 ± 0.037 |
| | 6435 | 0.040492 ± 0.000291 | 0.040509 ± 0.000288 | 0.04 ± 0.05 | 0.347 ± 0.031 |
| Wind Power | 1314 | 0.007335 ± 0.000262 | 0.007339 ± 0.000261 | 0.06 ± 0.10 | 0.501 ± 0.042 |
| | 2628 | 0.006363 ± 0.000130 | 0.006368 ± 0.000130 | 0.08 ± 0.05 | 0.468 ± 0.028 |
| | 3942 | 0.006580 ± 0.000098 | 0.006583 ± 0.000098 | 0.04 ± 0.04 | 0.493 ± 0.029 |
| | 5256 | 0.006583 ± 0.000084 | 0.006584 ± 0.000084 | 0.00 ± 0.03 | 0.498 ± 0.025 |
| | 6570 | 0.006175 ± 0.000050 | 0.006175 ± 0.000049 | -0.00 ± 0.02 | 0.528 ± 0.033 |
| Synthetic Regression | 200 | 0.023265 ± 0.000989 | 0.023317 ± 0.000979 | 0.29 ± 0.60 | 0.306 ± 0.029 |
| | 400 | 0.022073 ± 0.000552 | 0.022101 ± 0.000552 | 0.13 ± 0.26 | 0.365 ± 0.032 |
| | 600 | 0.021332 ± 0.000483 | 0.021358 ± 0.000490 | 0.12 ± 0.16 | 0.385 ± 0.039 |
| | 800 | 0.015924 ± 0.000382 | 0.015945 ± 0.000386 | 0.12 ± 0.09 | 0.381 ± 0.038 |
| | 1000 | 0.015908 ± 0.000365 | 0.015911 ± 0.000361 | 0.03 ± 0.12 | 0.419 ± 0.031 |
| Concrete Strength | 201 | 0.016621 ± 0.001047 | 0.016592 ± 0.001043 | -0.15 ± 0.28 | 0.362 ± 0.035 |
| | 402 | 0.017469 ± 0.000866 | 0.017431 ± 0.000886 | -0.33 ± 0.31 | 0.352 ± 0.031 |
| | 603 | 0.017323 ± 0.000472 | 0.017336 ± 0.000490 | 0.03 ± 0.25 | 0.406 ± 0.028 |
| | 804 | 0.016711 ± 0.000551 | 0.016726 ± 0.000563 | 0.06 ± 0.16 | 0.452 ± 0.028 |
| | 1005 | 0.015719 ± 0.000346 | 0.015722 ± 0.000348 | 0.01 ± 0.08 | 0.477 ± 0.024 |
| Energy Efficiency | 153 | 0.003620 ± 0.000316 | 0.003650 ± 0.000315 | 1.02 ± 0.92 | 0.413 ± 0.043 |
| | 306 | 0.002928 ± 0.000118 | 0.002926 ± 0.000118 | -0.03 ± 0.35 | 0.398 ± 0.038 |
| | 459 | 0.002872 ± 0.000120 | 0.002873 ± 0.000119 | 0.04 ± 0.22 | 0.453 ± 0.040 |
| | 612 | 0.002615 ± 0.000083 | 0.002621 ± 0.000081 | 0.29 ± 0.30 | 0.452 ± 0.035 |
| | 765 | 0.002667 ± 0.000054 | 0.002673 ± 0.000057 | 0.21 ± 0.19 | 0.436 ± 0.025 |
| House Price | 200 | 0.000103 ± 0.000006 | 0.000103 ± 0.000006 | 0.77 ± 0.52 | 0.344 ± 0.029 |
| | 400 | 0.000100 ± 0.000004 | 0.000101 ± 0.000004 | 0.10 ± 0.18 | 0.463 ± 0.037 |
| | 600 | 0.000100 ± 0.000004 | 0.000100 ± 0.000004 | 0.07 ± 0.12 | 0.515 ± 0.023 |
| | 800 | 0.000104 ± 0.000003 | 0.000104 ± 0.000003 | 0.03 ± 0.09 | 0.476 ± 0.046 |
| | 1000 | 0.000103 ± 0.000003 | 0.000104 ± 0.000003 | 0.22 ± 0.16 | 0.445 ± 0.040 |
| Parkinson's Monitoring | 1175 | 0.004750 ± 0.000108 | 0.004754 ± 0.000108 | 0.08 ± 0.10 | 0.425 ± 0.041 |
| | 2350 | 0.004672 ± 0.000093 | 0.004681 ± 0.000096 | 0.18 ± 0.14 | 0.333 ± 0.025 |
| | 3525 | 0.004651 ± 0.000093 | 0.004650 ± 0.000092 | -0.02 ± 0.06 | 0.355 ± 0.027 |
| | 4700 | 0.004618 ± 0.000074 | 0.004620 ± 0.000074 | 0.05 ± 0.05 | 0.449 ± 0.029 |
| | 5875 | 0.004655 ± 0.000053 | 0.004655 ± 0.000052 | 0.01 ± 0.03 | 0.429 ± 0.023 |
| Wine Quality | 1063 | 0.021526 ± 0.000636 | 0.021544 ± 0.000647 | 0.07 ± 0.23 | 0.393 ± 0.025 |
| | 2126 | 0.015126 ± 0.000310 | 0.015131 ± 0.000307 | 0.04 ± 0.05 | 0.452 ± 0.032 |
| | 3189 | 0.015448 ± 0.000199 | 0.015461 ± 0.000197 | 0.09 ± 0.05 | 0.443 ± 0.026 |
| | 4252 | 0.014830 ± 0.000261 | 0.014839 ± 0.000260 | 0.06 ± 0.04 | 0.487 ± 0.034 |
| | 5315 | 0.014894 ± 0.000162 | 0.014896 ± 0.000164 | 0.01 ± 0.03 | 0.505 ± 0.024 |

## H    STATISTICAL SIGNIFICANCE TESTS

For every dataset $\times$ training–set–fraction of our main experiment we did a 10-fold cross validation comparison of CRDA'S augmented MSE against the corresponding raw unaugmented MSE with a two–sided Wilcoxon signed–rank test ($n_{\text{folds}} = 10$, $n_{\text{seeds}} = 15$ per cell). The heat-maps in Figures 6 and 7 visualize the outcome.

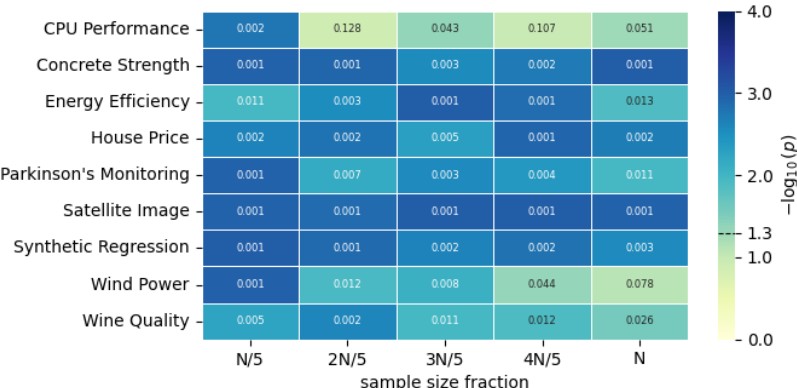

Figure 6: **MLP baseline.** Colour encodes $-\log_{10}(p)$; numbers are the mean $p$ across 15 seeds. The dashed line on the colour-bar marks the $\alpha = 0.05$ threshold ($-\log_{10} p \approx 1.3$).

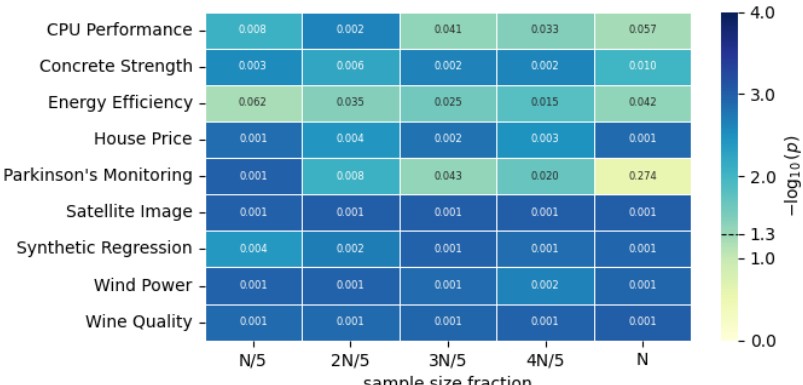

Figure 7: **XGB baseline.** Same layout and colour scale as Figure 6.

**Brief analysis.** Across *both* baselines the majority of cells are darker than the $\alpha = 0.05$ cut-off, indicating that CRDA delivers a statistically significant reduction in test–MSE for most dataset/-size combinations. Significance is strongest for smaller training sets and occasionally weakens as the full dataset is used (e.g. `CPU Performance` and `Wind Power` for MLP, `Parkinson's Monitoring` for XGB), but even at $n$ the method remains significant in 7/9 datasets with at least one baseline. These results support the robustness of the performance gains reported in the main paper.

## I    COMPLETE PER-DATASET SCORES

Table 10 reports the baseline MSE, CRDA MSE and their percentage change ($\Delta \%$) for every dataset and sample size subset. It is a more comprehensive version of Table 1 in the main paper. These results are the averages across 15 different seed runs and so we include their standard errors.[3]

---

[3]Per-seed results are available in the code repository at `experiments/{dataset}/{model}/interim_results`

Table 10: Complete results with standard errors for XGB and MLP across 15 seeds for each of the 9 datasets. Lower is better ↓.

| Dataset | Sample Size | XGB ↓ | | | MLP ↓ | | |
|---|---|---|---|---|---|---|---|
| | | $MSE_{baseline}$ | $MSE_{CRDA}$ | Δ % | $MSE_{baseline}$ | $MSE_{CRDA}$ | Δ % |
| CPU Performance | 1638 | $0.00097 \pm 0.00004$ | $0.00089 \pm 0.00004$ | -7.0 ± 2.7 | $0.00112 \pm 0.00007$ | $0.00087 \pm 0.00003$ | -20.2 ± 3.2 |
| | 3276 | $0.00088 \pm 0.00003$ | $0.00079 \pm 0.00002$ | -9.5 ± 2.2 | $0.00100 \pm 0.00002$ | $0.00085 \pm 0.00002$ | -14.0 ± 1.5 |
| | 4914 | $0.00077 \pm 0.00002$ | $0.00072 \pm 0.00001$ | -6.2 ± 1.2 | $0.00093 \pm 0.00002$ | $0.00082 \pm 0.00001$ | -11.3 ± 1.5 |
| | 6552 | $0.00073 \pm 0.00002$ | $0.00069 \pm 0.00001$ | -4.1 ± 1.6 | $0.00090 \pm 0.00003$ | $0.00079 \pm 0.00001$ | -10.5 ± 2.2 |
| | 8190 | $0.00074 \pm 0.00002$ | $0.00070 \pm 0.00001$ | -5.2 ± 1.9 | $0.00087 \pm 0.00001$ | $0.00078 \pm 0.00001$ | -10.2 ± 0.8 |
| Satellite Image | 1287 | $0.01778 \pm 0.00046$ | $0.01697 \pm 0.00051$ | -4.5 ± 1.4 | $0.02031 \pm 0.00100$ | $0.01629 \pm 0.00057$ | -18.4 ± 3.1 |
| | 2574 | $0.01636 \pm 0.00035$ | $0.01576 \pm 0.00040$ | -3.7 ± 0.8 | $0.01747 \pm 0.00037$ | $0.01455 \pm 0.00039$ | -16.7 ± 1.5 |
| | 3861 | $0.01460 \pm 0.00034$ | $0.01390 \pm 0.00034$ | -4.8 ± 0.8 | $0.01585 \pm 0.00053$ | $0.01211 \pm 0.00035$ | -23.1 ± 1.7 |
| | 5148 | $0.01366 \pm 0.00032$ | $0.01300 \pm 0.00030$ | -4.7 ± 1.1 | $0.01415 \pm 0.00044$ | $0.01076 \pm 0.00029$ | -23.7 ± 1.2 |
| | 6435 | $0.01254 \pm 0.00029$ | $0.01186 \pm 0.00026$ | -5.3 ± 0.8 | $0.01232 \pm 0.00034$ | $0.00989 \pm 0.00028$ | -19.7 ± 1.0 |
| Wind Power | 1314 | $0.00742 \pm 0.00028$ | $0.00721 \pm 0.00028$ | -2.8 ± 1.2 | $0.00752 \pm 0.00024$ | $0.00697 \pm 0.00024$ | -7.2 ± 1.5 |
| | 2628 | $0.00602 \pm 0.00012$ | $0.00603 \pm 0.00012$ | 0.2 ± 0.6 | $0.00621 \pm 0.00016$ | $0.00562 \pm 0.00011$ | -9.2 ± 1.4 |
| | 3942 | $0.00586 \pm 0.00008$ | $0.00578 \pm 0.00008$ | -1.3 ± 0.4 | $0.00593 \pm 0.00007$ | $0.00539 \pm 0.00008$ | -9.0 ± 0.7 |
| | 5256 | $0.00570 \pm 0.00006$ | $0.00562 \pm 0.00007$ | -1.4 ± 0.4 | $0.00567 \pm 0.00008$ | $0.00533 \pm 0.00008$ | -6.2 ± 0.4 |
| | 6570 | $0.00528 \pm 0.00005$ | $0.00522 \pm 0.00005$ | -1.1 ± 0.3 | $0.00530 \pm 0.00004$ | $0.00500 \pm 0.00004$ | -5.6 ± 0.5 |
| Synthetic Regression | 200 | $0.00652 \pm 0.00043$ | $0.00564 \pm 0.00031$ | -12.0 ± 3.7 | $0.01993 \pm 0.00172$ | $0.01387 \pm 0.00157$ | -28.8 ± 6.3 |
| | 400 | $0.00327 \pm 0.00026$ | $0.00312 \pm 0.00022$ | -3.2 ± 2.5 | $0.00610 \pm 0.00036$ | $0.00384 \pm 0.00031$ | -36.9 ± 3.0 |
| | 600 | $0.00264 \pm 0.00008$ | $0.00242 \pm 0.00007$ | -7.9 ± 2.2 | $0.00321 \pm 0.00026$ | $0.00228 \pm 0.00019$ | -27.9 ± 3.0 |
| | 800 | $0.00165 \pm 0.00008$ | $0.00161 \pm 0.00009$ | -2.2 ± 2.0 | $0.00223 \pm 0.00016$ | $0.00140 \pm 0.00008$ | -34.1 ± 4.0 |
| | 1000 | $0.00152 \pm 0.00005$ | $0.00145 \pm 0.00006$ | -4.6 ± 2.8 | $0.00220 \pm 0.00013$ | $0.00123 \pm 0.00007$ | -42.3 ± 3.1 |
| Concrete Strength | 201 | $0.00777 \pm 0.00068$ | $0.00701 \pm 0.00063$ | -8.0 ± 3.6 | $0.01033 \pm 0.00103$ | $0.00793 \pm 0.00050$ | -17.8 ± 5.7 |
| | 402 | $0.00493 \pm 0.00035$ | $0.00453 \pm 0.00037$ | -8.4 ± 2.7 | $0.00635 \pm 0.00053$ | $0.00496 \pm 0.00037$ | -19.8 ± 2.8 |
| | 603 | $0.00473 \pm 0.00024$ | $0.00427 \pm 0.00024$ | -9.7 ± 2.2 | $0.00602 \pm 0.00014$ | $0.00494 \pm 0.00014$ | -17.6 ± 2.5 |
| | 804 | $0.00365 \pm 0.00017$ | $0.00307 \pm 0.00014$ | -15.7 ± 1.9 | $0.00497 \pm 0.00026$ | $0.00361 \pm 0.00013$ | -24.8 ± 4.1 |
| | 1005 | $0.00290 \pm 0.00010$ | $0.00256 \pm 0.00013$ | -12.2 ± 2.0 | $0.00422 \pm 0.00024$ | $0.00306 \pm 0.00016$ | -26.9 ± 1.7 |
| Energy Efficiency | 153 | $0.00399 \pm 0.00048$ | $0.00344 \pm 0.00050$ | -13.3 ± 7.7 | $0.00583 \pm 0.00048$ | $0.00426 \pm 0.00046$ | -25.1 ± 6.8 |
| | 306 | $0.00233 \pm 0.00014$ | $0.00206 \pm 0.00015$ | -12.2 ± 3.1 | $0.00321 \pm 0.00014$ | $0.00233 \pm 0.00021$ | -28.1 ± 4.8 |
| | 459 | $0.00165 \pm 0.00012$ | $0.00143 \pm 0.00011$ | -10.5 ± 5.9 | $0.00188 \pm 0.00015$ | $0.00106 \pm 0.00013$ | -43.0 ± 4.6 |
| | 612 | $0.00128 \pm 0.00007$ | $0.00100 \pm 0.00006$ | -19.3 ± 5.3 | $0.00091 \pm 0.00008$ | $0.00052 \pm 0.00005$ | -40.7 ± 3.9 |
| | 765 | $0.00097 \pm 0.00006$ | $0.00076 \pm 0.00007$ | -21.0 ± 4.4 | $0.00053 \pm 0.00008$ | $0.00035 \pm 0.00003$ | -28.3 ± 4.3 |
| House Price | 200 | $0.00079 \pm 0.00008$ | $0.00064 \pm 0.00005$ | -14.2 ± 4.8 | $0.00102 \pm 0.00011$ | $0.00057 \pm 0.00007$ | -40.6 ± 4.8 |
| | 400 | $0.00033 \pm 0.00002$ | $0.00031 \pm 0.00002$ | -5.4 ± 2.3 | $0.00041 \pm 0.00003$ | $0.00025 \pm 0.00001$ | -37.0 ± 3.7 |
| | 600 | $0.00027 \pm 0.00002$ | $0.00026 \pm 0.00002$ | -4.9 ± 2.7 | $0.00029 \pm 0.00002$ | $0.00020 \pm 0.00002$ | -30.1 ± 3.8 |
| | 800 | $0.00024 \pm 0.00001$ | $0.00022 \pm 0.00001$ | -9.9 ± 2.0 | $0.00023 \pm 0.00001$ | $0.00016 \pm 0.00001$ | -30.3 ± 4.1 |
| | 1000 | $0.00020 \pm 0.00001$ | $0.00018 \pm 0.00001$ | -6.5 ± 1.9 | $0.00019 \pm 0.00001$ | $0.00014 \pm 0.00001$ | -27.0 ± 2.5 |
| Parkinson's Monitoring | 1175 | $0.00079 \pm 0.00003$ | $0.00072 \pm 0.00003$ | -8.4 ± 2.4 | $0.00165 \pm 0.00012$ | $0.00101 \pm 0.00006$ | -36.2 ± 3.9 |
| | 2350 | $0.00034 \pm 0.00002$ | $0.00032 \pm 0.00001$ | -6.6 ± 2.8 | $0.00080 \pm 0.00005$ | $0.00054 \pm 0.00003$ | -31.8 ± 2.5 |
| | 3525 | $0.00021 \pm 0.00001$ | $0.00020 \pm 0.00001$ | -2.8 ± 3.4 | $0.00048 \pm 0.00003$ | $0.00030 \pm 0.00002$ | -36.6 ± 4.0 |
| | 4700 | $0.00015 \pm 0.00001$ | $0.00014 \pm 0.00001$ | -6.3 ± 2.4 | $0.00042 \pm 0.00003$ | $0.00021 \pm 0.00001$ | -46.4 ± 4.1 |
| | 5875 | $0.00011 \pm 0.00001$ | $0.00011 \pm 0.00001$ | 1.7 ± 3.8 | $0.00026 \pm 0.00002$ | $0.00013 \pm 0.00001$ | -47.2 ± 4.6 |
| Wine Quality | 1063 | $0.02057 \pm 0.00056$ | $0.02062 \pm 0.00054$ | 0.3 ± 0.8 | $0.02291 \pm 0.00088$ | $0.02284 \pm 0.00129$ | -0.3 ± 3.3 |
| | 2126 | $0.01416 \pm 0.00029$ | $0.01429 \pm 0.00029$ | 1.0 ± 0.7 | $0.01539 \pm 0.00026$ | $0.01458 \pm 0.00032$ | -5.2 ± 1.6 |
| | 3189 | $0.01391 \pm 0.00019$ | $0.01386 \pm 0.00016$ | -0.3 ± 0.5 | $0.01478 \pm 0.00023$ | $0.01423 \pm 0.00024$ | -3.6 ± 1.5 |
| | 4252 | $0.01332 \pm 0.00024$ | $0.01324 \pm 0.00026$ | -0.6 ± 0.4 | $0.01386 \pm 0.00027$ | $0.01323 \pm 0.00025$ | -4.4 ± 0.8 |
| | 5315 | $0.01332 \pm 0.00012$ | $0.01318 \pm 0.00014$ | -1.1 ± 0.3 | $0.01397 \pm 0.00016$ | $0.01328 \pm 0.00019$ | -5.0 ± 0.6 |

## J  ADDITIONAL BASELINE: CATBOOST ANALYSIS

To assess CRDA's robustness against stronger tree-based ensembles, we performed an additional evaluation using CatBoost (Prokhorenkova et al., 2018). CatBoost is often considered better than XGBoost due to its oblivious trees and robustness to overfitting, making it a challenging predictor to improve upon.

We evaluated performance at three fixed sample sizes ($N = \{300, 500, 700\}$) to observe behavior across different data availabilities.

Table 11 presents the percentage change in MSE (Δ%). We observe three distinct behaviors:

- **Consistent Gains:** On *House Price* and *Wind Power*, CRDA significantly reduces MSE across all sample sizes (peaking at -22.8% for *House Price*), demonstrating that CRDA behaves robustly for these tasks regardless of sample size.
- **Late-Stage Gains:** *CPU Performance* requires a sufficient number of samples to model the residual. It shows no benefit at $N = 300$ but improves substantially as data increases, reaching -13.0% at $N = 700$.
- **Sweet-Spot Behavior:** Datasets such as *Parkinson's Monitoring*, *Energy Efficiency*, and *Synthetic Regression* exhibit a "sweet spot" around $N = 500$, where the augmentation provides the most benefit ($\approx$ 4-5% reduction) before CatBoost potentially saturates the signal at larger sample sizes.

Table 11: Percentage change in MSE ($\Delta\%$) for *CatBoost* at fixed sample sizes. Values represent the mean $\Delta\%$ across 15 seeds $\pm$ standard error.

| Dataset | $N = 300$ | $N = 500$ | $N = 700$ |
|---|---|---|---|
| House Price | -22.80 $\pm$ 0.60 | -18.87 $\pm$ 0.52 | -14.11 $\pm$ 0.50 |
| CPU Performance | 1.92 $\pm$ 0.91 | -7.34 $\pm$ 0.58 | -13.04 $\pm$ 0.90 |
| Parkinson's Monitoring | -1.44 $\pm$ 0.61 | -5.13 $\pm$ 0.60 | -1.33 $\pm$ 0.46 |
| Energy Efficiency | -1.65 $\pm$ 0.86 | -4.95 $\pm$ 1.01 | -1.89 $\pm$ 0.90 |
| Synthetic Regression | -2.31 $\pm$ 0.57 | -4.01 $\pm$ 0.60 | -1.15 $\pm$ 0.66 |
| Wind Power | -3.78 $\pm$ 0.38 | -2.54 $\pm$ 0.24 | -2.47 $\pm$ 0.29 |
| Satellite Image | -0.95 $\pm$ 0.53 | -2.08 $\pm$ 0.41 | -1.95 $\pm$ 0.30 |
| Wine Quality | -0.09 $\pm$ 0.39 | -1.62 $\pm$ 0.29 | -1.94 $\pm$ 0.21 |
| Concrete Strength | -0.95 $\pm$ 0.56 | -0.69 $\pm$ 0.39 | 0.46 $\pm$ 0.39 |

## K  COMPUTE BUDGET AND CARBON FOOTPRINT

- **Hardware.** AWS `c7i.24xlarge` (96 vCPU, 192 GB RAM, Xeon Platinum 8480C, $\approx$0.59 kW active draw).[4]

- **Runtime.** 13.562 h total (9.103 h MLP baseline, 4.459 h XGB baseline).

- **Energy.** $13.562 \times 0.59 \approx 8.0$ kWh.

- **CO$_2$-eq.** Local grid intensity 34.5 g $CO_2$/kWh $\Rightarrow 8.0 \times 0.0345 \approx 0.28$ kg $CO_2$.

## L  BROADER SOCIETAL IMPACT CONSIDERATIONS

This is foundational work that aims to improve regression in scarce data scenarios. As discussed in Section 7 (Limitations) of the main paper, CRDA could worsen predictive accuracy instead of improving it, leading to negative consequences in high impact applications. To mitigate such negative outcomes, CRDA filters applications with the PC algorithm, the Pearson correlation test and the Wilcoxon signed rank test.

## M  PROOF OF PROPOSITION 1

**Recall Assumption 1:**

Let the feature vector $X$ be partitioned into two disjoint subsets, $X = (X_P, X_R)$, where $X_P$ are the features we intend to perturb (the *perturbable* coordinates) and $X_R$ are the features we hold fixed (the *remaining* coordinates). Let $g(X) = \mathbb{E}[Y|X]$ be the true conditional expectation function, and let $Z = Y - g(X)$ be the corresponding structural noise term. We introduce the following condition:

$$\mathbb{P}(Z \mid X_P, X_R) = \mathbb{P}(Z \mid X_R) \tag{2}$$

Equation 2 says that the noise $Z$ is conditionally independent of the perturbable features $X_P$ given the fixed features $X_R$.

**Proposition 1 stated:**

Suppose Assumption 1 holds. Then for any $x_R$ in the support of $X_R$ and any $x_P, x'_P$ in the conditional support of $X_P \mid X_R = x_R$, we have

$$\mathbb{P}(Z \mid X_P = x_P, X_R = x_R) = \mathbb{P}(Z \mid X_P = x'_P, X_R = x_R).$$

Equivalently, $\mathbb{P}(Z \mid X_P = x_P, X_R) = \mathbb{P}(Z \mid X_R)$ is constant in $x_P$.

---

[4]Power estimate from Intel C7i workload proof sheet.

*Proof.* By Assumption 1, there exists a version of the regular conditional law such that, for (almost) [5] every $(x_P, x_R)$ and every measurable set $A$,

$$\mathbb{P}\big(Z \in A \mid X_P = x_P, X_R = x_R\big) \;=\; \mathbb{P}\big(Z \in A \mid X_R = x_R\big)$$

Fix $x_R$ and two values $x_P, x'_P$ in the conditional support of $X_P \mid X_R = x_R$. Applying the displayed equality once with $x_P$ and once with $x'_P$ yields

$$\mathbb{P}\big(Z \in A \mid X_P = x_P, X_R = x_R\big) \;=\; \mathbb{P}\big(Z \in A \mid X_R = x_R\big) \;=\; \mathbb{P}\big(Z \in A \mid X_P = x'_P, X_R = x_R\big),$$

for all measurable $A$. Hence the conditional laws coincide, proving the claim. $\qquad\square$

---

[5]Conditional distributions are defined only up to sets of probability zero, so equalities hold almost surely. We also restrict $x_P, x'_P$ to the conditional support of $X_P \mid X_R = x_R$ (positivity) to ensure the displayed conditionals are well-defined.

