# OpenReview forum: "Counterfactual Residual Data Augmentation for Regression"
_ICLR.cc/2026/Conference — ICLR 2026 Conference Desk Rejected Submission_

### Official Review · Reviewer_qFtg · 2025-10-19

**Soundness:** 3
**Presentation:** 4
**Contribution:** 3
**Rating:** 6
**Confidence:** 4

**Summary:**

This paper proposes a data augmentation technique called as counterfactual residual data augmentation (CRDA) for tabular regression problem. When the number of samples are few, CRDA empirically provides evidence of performance improvements. CRDA relies on residual invariance assumption and the experiments show improvements over baselines.

**Strengths:**

1. The paper is written well, easy to read and understand.
2. The proposed method is simple and model agnostic.
3. This paper addresses crucial challenges in real-world tabular tasks such as overfitting and limited sample size.
3. Experimental results show good improvements compared to baselines.

**Weaknesses:**

1. There is no theoretical analysis of how the proposed method achieves good performance using the proposed data augmentation method.
2. CRDA is applied on only two methods: MLP and XGBoost. For example, CatBoost often outperforms XGBoost. It is crucial to check if other popular methods for tabular data can benefit from the CRDA.
3. What kinds of underlying causal data generating processes or causal graphs satisfy the Assumption 1? I believe the problems caused by hidden confounding (see [1-2] below), which is common in real-world tabular data, cannot be addressed by CRDA. I appreciate some discussion around it both in rebuttal and in the revision.
4. In step 3 of the algorithm, if a causal graph check is performed using PC algorithm, it is mentioned that all variables that are directly connected to the residual are excluded from $X_P$ but if a correlation check is performed, variables with high correlation are excluded. These two ideas contradict with each other in a case where there is a variable that is highly correlated with $Z$ but is indirectly connected to $Z$.
5. As acknowledged, when the dataset size is small, obtaining reliable estimates for causal graph or statical tests is challenging (step 3 of the algorithm).

References:

1. Prashanth et al. Scalable Out-of-distribution Robustness in the Presence of Unobserved Confounders
2. Gowtham et al. When Shift Happens - Confounding Is to Blame.

**Questions:**

See the weaknesses section.

---

> ### Author Response · Authors · 2025-11-25
> **Rebuttal by Authors**
>
> ## Rebuttal
>
> We thank the reviewer for the careful reading and the constructive questions.
> Below we respond point-by-point (tags **W2–W5** for weaknesses).
>
> ### 1 Limited Model Selection (W2)
>
> We focused on XGBoost and MLPs to represent two widely used and complementary baselines in tabular learning: boosted-tree ensembles and neural networks. However, we agree that CatBoost is a state-of-the-art predictor that can often produce stronger results. To address the concern, we are currently running additional experiments using CatBoost. As soon as the results are done we will add them in the revision pdf and report here with a new comment.
>
> ### 2 Hidden Confounding (W3)
>
> We appreciate the references regarding hidden confounding and agree that it is a primary cause of distribution shift failures. In the case of CRDA, if an unobserved confounder $U$ affects both a feature $X_P$ and the outcome $Y$, it induces a dependence between $X_P$ and the residual term $Z$ (since $Z$ implicitly captures the variation from $U$). This would violate our Assumption 1.
>
> Regarding what causal graph would satisfy Assumption 1, $P(Z | X_P, X_R) = P(Z | X_R)$, it would have to be an Additive Noise Model (ANM) where the features in $X_P$ are exogenous to the noise mechanism $Z$. Formally, this corresponds to a causal graph where there is no open "backdoor path" between $X_P$ and $Y$ that passes through $Z$ (i.e., no unobserved common cause $U \to X_P$ and $U \to Y$).
>
> Crucially, CRDA handles confounding via detection, not assumption.
> If hidden confounding exists ($U \to X_P, U \to Y$), it renders $X_P$ and $Z$ statistically dependent. Step 3 of our algorithm (the Correlation and PC checks) is designed specifically to detect this signal. If a feature $X_P$ carries a strong correlation with the residual $Z$ (which acts as a proxy for the hidden confounder $U$, as noted in [2]), CRDA rejects it from the perturbation set and moves it to $X_R$. Therefore, CRDA is conservative: it limits augmentation to the subset of features where the "no-confounding" condition empirically holds.
>
> We have updated the paper in Section 3.2 [Lines 169-194] to explicitly show the causal structure and discuss the implication of these references.
>
> ### 3 Contradiction in Step 3? (W4)
>
> We apologize for any confusion in Step 3 of our algorithm. We do not treat the PC and correlation checks as alternatives, but as **two conservative filters applied in sequence**.
> We first run the PC algorithm and *exclude* any feature adjacent to $Z$ in the learned graph; we then further *exclude* any remaining feature whose Pearson correlation with $Z$ exceeds a threshold.
>
> Thus, a feature is included in $X_P$ only if (i) it is non-adjacent to $Z$ in the PC graph **and** (ii) it has low empirical correlation with $Z$.
>
> We hope this clarifies that there is *no contradiction* and shows how a variable that is highly correlated with $Z$ but indirectly connected in the graph, will still be removed from $X_P$ by the correlation test.
>
>
> ### 4 Small Sample Size Reliability (W5)
>
> You are correct that causal discovery and independence tests lose power in small-sample regimes. This is precisely why CRDA includes the Wilcoxon Signed-Rank Test (Step 6) as a final "safety gate."
>
> We view Step 3 (Feature Selection) as a heuristic to propose candidate augmentations. Step 6 then empirically validates if those candidates actually aid generalization. As shown in our sample-size scaling experiments (Fig. 3) and the Linear Regression study (Appendix G), when the data is too sparse to estimate residuals or independence correctly, the method fails to find significant improvement and defaults to the baseline, ensuring no harm is done.

---

> > ### Comment · Reviewer_qFtg · 2025-11-25
> > **Thank you for the response**
> >
> > I thank the authors for providing detailed responses. My concerns are addressed except W1. The paper provides a simple and effective method for data augmentation. Hence, I maintain my positive assessment of the paper. I encourage the authors to add the new results on CatBoost in the revision.

---

> ### Author Response · Authors · 2025-11-27
> **Update on W2**
>
> We have now updated the paper with the results for CatBoost as a base model.
>
> We conducted an evaluation of CatBoost across all 9 benchmark datasets using 15 random seeds. We evaluated performance at three distinct sample sizes ($N=300, 500, 700$) to characterize behavior across data regimes. These results have been added to Appendix J [Lines 1168-1200].
>
> As anticipated, CatBoost proved to be a highly resilient baseline. However, CRDA still demonstrated the ability to extract significant signal even from this strong predictor.
>
> We made 3 key observations:
> 1. Robust Gains: On *House Price*, CRDA reduced MSE by 14-23% across all sample sizes, demonstrating that the CRDA was beneficial for this task even against a strong CatBoost baseline.
> 2. Regime-Specific Sweet Spots: We observed that optimal augmentation sometimes depends on the dataset manifold. For *CPU Performance*, CRDA requires a minimum number of samples to learn the residual structure, transitioning from neutral at $N=300$ to a 13.0% reduction at $N=700$. Conversely, *Parkinson's Monitoring*, *Energy Efficiency*, and *Synthetic Regression* showed a distinct "sweet spot" at $N=500$.
> 3. Neutral Cases: In cases where CatBoost was strong and left no exploitable signal (e.g., *Wine Quality*, *Concrete Strength*), significant performance degradation was not observed.
>
> We hope these results strengthen our paper and thank you for your positive assessment.

---

### Official Review · Reviewer_rEDf · 2025-10-29

**Soundness:** 2
**Presentation:** 3
**Contribution:** 3
**Rating:** 6
**Confidence:** 4

**Summary:**

The paper proposes a Counterfactual Residual Data Augmentation (CRDA) method, increasing the dataset size while remaining loyal to the residual distribution under the fixed identified invariant feature set. The proposed method is shown to improved generalization in regression tasks over a range of tabular datasets, mitigating the overfitting problem in small-dataset settings.

**Strengths:**

Overfitting to small datasets is a well-known problem in regression and domain-agnostic data augmentation is not straight forward due to inability to find the transformations that retains the correct labels. The motivation and problem definition is clearly stated in the paper. The authors provide a brief background summary, propose a novel method to augment the residual noise by identifying and counterfactually perturbing the invariant feature set and use this procedure to generate new sample points that preserve the noise structure and remain consistent with the systematic behavior of the dataset.

The proposed CRDA method improves the prediction accuracy and reduces the variance in regression tasks with tabular data especially in the small-dataset regime.

The paper is well written and the reasoning is easy to follow. The theoretical analysis is supported by the results in the experiments section. The idea is to incorporates residual with novelty to perform data augmentation, while residuals are good identifiers for model uncertainty.

**Weaknesses:**

While data augmentation for regression is a scarcely studied complex problem, some previous work are missing from the paper e.g. [Hwang'21, Schneider'23] which might be relevant to include (not relevant to the given rating).

The empirical results in the paper are limited to comparison with no-augmentation baseline on UCI and comparison with generative augmentation on synthetic 3d dataset. It is difficult to assess the true potential of the method when it is not compared to other regression data augmentation algorithms like RegMix, C-Mix or Anchor data augmentation on the more complex sets of datasets. I will reconsider the given rating if e.g. UCI experiments with any of these algorithms are included in the results.

[Hwang'21] Regmix: Data mixing augmentation for regression, Seong-Hyeon Hwang, Steven Euijong Whang, arXiv preprint arXiv:2106.03374, 2021

[Schneider'23] Anchor Data Augmentation,Nora Schneider, Shirin Goshtasbpour, Fernando Pérez-Cruz, NeurIPS 2023

**Questions:**

Similar to the proposed CRDA method, Anchor Data Augmentation (ADA) [Schneider'22] is rooted in CSM and linear augmentation of the features. The non-linear version of the augmentation process is the most similar to CRDA although in ADA augmentation alters the labels which is estimated according to the first-order Taylor approximation of the regressor. CRDA comes at the additional cost of Peter-Clark and correlation based feature elimination to retain the original labels of the data and relies on the assumption that removing the first order and linear connections to the residual is sufficient to retain the labels.

1 - Can the authors explain whether the feature selection phase in CRDA can be reformulated as an improved instance of Anchor matrix selection in a pretraining phase in ADA? If not how different the generated augmentations in CRDA are from ADA's in theory?

2 - What is the additional computation complexity of the method due to the feature selection process?

3 - Can the authors show how realistic the Assumption 1 is in practice e.g. for UCI? What is the expected divergence between $\mathbb P(Z|X_P,X_R)$ and $\mathbb P(Z|X_R)$ ? How effective is Peter-Clark or correlation removal step in reducing the divergence?

---

> ### Author Response · Authors · 2025-11-25
> **Rebuttal by Authors (Part 1)**
>
> ## Rebuttal (Part 1)
>
> We thank the reviewer for the careful reading and the constructive questions.
> Below we respond point-by-point (tags **W** for weaknesses and **Q1–Q3** for questions).
>
> ### 1 Missing Work (W)
>
> We thank the reviewer for pointing out and identifying the missing literature. We have updated the Related Work section to discuss RegMix, C-Mixup and Anchor Data Augmentation (ADA) [Lines 73-84].
>
> As per the request, we have performed the suggested comparisons and have extended our evaluation to include ADA and C-Mixup (Note: We were unable to include RegMix as no code release is available).
> To avoid confusion, we would also like to point out that Table 2 in the submitted manuscript explicitly compares baselines on all 9 real-world datasets (UCI/PMLB). The synthetic 3D dataset was only for the sample size scaling experiment (now shown as Fig 3). The new results follow the same setup and are run on the same 9 datasets (UCI/PMLB).
>
> Results (see updated Table 2 [Lines 428-460]):
> The new results highlight a critical trade-off: while ADA and C-Mixup can perform very well on specific tasks (e.g., House Price), they suffer from catastrophic failure modes on others.
> - Instability of ADA/C-Mixup: On the Parkinson’s dataset, ADA increases test error by +19.5% (MLP) and +89.5% (XGB), while C-Mixup explodes to +102.6% and +105.1%. Similarly, on Synthetic Regression, both methods significantly degrade performance.
> - Robustness of CRDA: In contrast, CRDA reduces error on Parkinson’s by -51.1% and -0.3%. CRDA also achieves the lowest MSE in the majority of cases. Even in cases where C-Mixup/ADA performs well (e.g., House Price), CRDA remains competitive without the risk of massive error spikes seen in geometric and generative augmentation methods.
>
> We kindly ask you to reconsider your score in light of these new results.
>
> ### 2 CRDA vs. Anchor Data Augmentation (ADA) (Q1)
>
> Regarding whether CRDA’s feature selection is an improved instance of ADA’s anchor matrix selection, we examined the pretraining phase for ADA and have concluded our method is distinct in two fundamental ways:
>
> 1. **Selection Mechanism:** ADA’s pretraining phase typically involves k-means clustering to construct an Anchor Matrix $A$ that identifies local homogeneous groups. CRDA, conversely, uses the PC Algorithm to identify a feature subset $X_P$ that is globally conditionally independent of the residual $Z$.
> 2. **Generative Assumption:**
> ADA relies on a first-order Taylor approximation (tangent plane) to generate labels, effectively assuming local linearity within anchor groups and so can fail when the curve is complex/sharp. CRDA makes no linearity assumption on the predictor function $g(x)$. Instead, we assume residual invariance ($Z \perp X_P | X_R$). This allows CRDA to generate valid counterfactuals (new labels) even in highly non-linear regions of the manifold, provided the noise structure is preserved.
>
> Therefore, CRDA is not a special case of ADA but a complementary approach.

---

> > ### Author Response · Authors · 2025-11-25
> > **Rebuttal by Authors (Part 2)**
> >
> > ## Rebuttal (Part 2)
> >
> > ### 3 Computational Complexity (Q2
> >
> > The feature selection consists of two steps:
> > 1. Pearson Correlation: This step scales linearly as $O(N \cdot d)$, where $N$ is the sample size and $d$ is the number of features.
> > 2. PC Algorithm: While the PC algorithm has a worst-case complexity that is exponential in $d$, it runs in polynomial time, roughly $O(N \cdot d^q)$, for sparse graphs (where $q$ is the maximum degree of the graph).
> >
> > Thus, the computational complexity of the feature selection process is dominated by the PC algorithm, which we use to create a causal graph and identify features with no links to Z.
> >
> > Practical Implications: In the context of tabular regression, the feature dimensionality $d$ is typically small (in our benchmarks, $7 \le d \le 36$). Due to this bounded dimensionality, the exponential worst-case of the PC algorithm is not always realized in practice.
> >
> > We measured the wall-clock time for the feature selection step (Algorithm 1, Step 3) on our 2 largest datasets in terms of $N$ and $d$
> >
> > (CPU Performance, $N=8192$ & $d=12$)
> > Here we found it takes approximately 0.41 seconds on a standard CPU.
> >
> > (Satellite Image, $N=6435$ & $d=36$)
> > Here we found it takes approximately 2.7 seconds on a standard CPU.
> >
> > ### 4 Validity of Assumption 1 & Divergence (Q3)
> > To empirically quantify the divergence between $P(Z|X_P,X_R)$ and $P(Z|X_R)$, we utilized Mutual Information (MI). MI is formally equivalent to the KL-Divergence between the joint distribution and the product of marginals: $I(Z; X) = D_{KL}(P(Z, X) || P(Z)P(X))$. A value of zero indicates perfect independence.
> >
> > We conducted an analysis across all 9 datasets (15 random seeds, XGBoost baseline), estimating MI using the Kraskov (KSG) estimator. We compared the features Selected by CRDA ($X_P$) against those Rejected by our PC/Correlation filter ($X_R$).
> >
> > We observed two distinct behaviors (*full results in Appendix F.1 Table 7* [Lines 967-997]):
> >
> > 1. High Discrimination Scenarios: For datasets where specific features violated the independence assumption, our filter successfully identified and rejected them. For example on *Energy Efficiency*, rejected features had 2.9x higher divergence ($0.0160$ nats) than selected features ($0.0054$ nats). On *House Prices*, rejected features had 2.8x higher divergence ($0.0155$ nats) than selected features ($0.0056$ nats).
> > 2. Natural Independence Scenarios: In datasets like Wine Quality and Wind Power, the MI was negligible ($<0.013$) for both selected and rejected sets. This indicates that the independence assumption $Z \perp X$ holds naturally for the majority of features in these domains, and CRDA correctly proceeds without aggressive filtering.
> >
> > This analysis thus confirms that the feature selection step effectively identifies the subset of features where the conditional independence assumption holds.

---

### Official Review · Reviewer_S8VY · 2025-11-01

**Soundness:** 2
**Presentation:** 2
**Contribution:** 3
**Rating:** 4
**Confidence:** 3

**Summary:**

The paper proposes Counterfactual Residual Data Augmentation (CRDA) for improving the performance of regression on tabular data. It is based on the assumption that there some features on which perturbations do not alter the distribution of the residual given a reasonably good regression model. After fitting a base regressor, the method identifies “perturbable” features that are conditionally independent of the residual given the remaining features, and generates counterfactual inputs by applying small perturbations only on those features. The counterfactual labels are generated by adding the predicted value of the base model and the original residual. Across nine benchmarks, CRDA reduces MSE for MLPs by 22.9% and for XGBoost by 6.4%, and outperforms TabDDPM, TVAE, and CTGAN on average

**Strengths:**

* A key strength of this work is that it tackles an important yet under-explored problem: principled data augmentation for tabular regression. Unlike many tasks in computer vision and NLP, augmenting tabular data with counterfactual regression labels is less obvious, especially compared to classification where the labels are discrete and finite. The paper addresses this gap by formalizing a residual-invariance assumption and prove its viability through empirical experiments.
* The core idea of the proposed method is easy to implement. After identifying the features, the augmentation process is simply adding noise and re-compute the labels. The feature selection process is flexible and can be performed according to various criteria.
* The experimental evidence is strong. The approach consistently reduces improves performance across nine tabular-regression benchmarks for both MLP and XGBoost.

**Weaknesses:**

* As discussed at the end of Section 6, the method does not work well if the dataset is too small because it needs a reasonably good model to begin with.
* Some simple and simple baselines/ablations are missing, e.g. perturbing all features randomly, fixing the label after perturbation, etc. These experiments will help verify the basic assumption of residual invariant to some features.
* The intuition behind the assumption still doesn’t convince me.

**Questions:**

* Is it possible to do this iteratively (train a model with augmented data, and then generate new augmentation from the new model) to get further improvement?

---

> ### Author Response · Authors · 2025-11-25
> **Rebuttal by Authors**
>
> ## Rebuttal
>
> We thank the reviewer for the careful reading and the constructive questions.
> Below we respond point-by-point (tags **W1–W3** for weaknesses, **Q1** for the question).
>
> ### 1 Small Data Settings (W1)
> We agree with the reviewer that CRDA is not intended for extremely small datasets where the base regressor cannot meaningfully separate signal from noise. Our goal is to improve performance in the practical "sweet spot" where:
> (i) the baseline model has captured a non-trivial portion of the signal, but (ii) data is still scarce enough that additional local exploration in feature space is beneficial.
>
> Empirically, our synthetic sample-size experiment (Fig. 3) already illustrates this behaviour. CRDA brings the largest MSE reductions between ≈2.5k and 20k samples, while offering less benefit at the two extreme ends of the sample sizes. In Appendix G we further study a deliberately weak base predictor (linear regression). In every dataset/fold, the Wilcoxon gate yields p ≥ α, so CRDA abstains and reverts to the baseline model, displaying the effectiveness of our safegaurds against performance degradation.
>
> ### 2 Simple Ablations (W2)
>
> We appreciate the suggestion to verify our assumptions through simpler baselines. We conducted ablation studies on three representative datasets (Synthetic Regression, Energy Efficiency, and Parkinson's Monitoring) using both MLP and XGBoost as the base regressors. We compared CRDA against:
>
> 1. Global Perturbation: Perturbing all features randomly without independence checks.
> 2. Label Invariance: Perturbing features but keeping the label $y$ fixed.
>
> The results (*now added in Appendix F.2 Table 8* [Lines 1001-1025]) confirm that CRDA consistently outperforms these baselines. For example, on the Parkinson's dataset with an MLP, CRDA reduces MSE by 58.4%, whereas Global Perturbation only achieves a 13.5% reduction, and Label Invariance actually degrades performance (+0.5%). Similarly, for XGBoost on Synthetic and Parkinson's, Global Perturbation increases the MSE, confirming that indiscriminate perturbation can be harmful and validating the benefit of CRDA's residual-invariance feature selection.
>
> ### 3 Unintuitive Assumption (W3)
>
> In most regression models, the data-generating process is represented as a standard additive-noise model:
> $$
> Y = g(X) + Z
> $$
> where $g(X) = E[Y|X]$ is the systematic component and $Z$ represents unobserved factors. A common modelling assumption is that, once $g$ is well specified, then $Z$ is independent or at least uncorrelated to $X$. Our Assumption 1 is strictly *weaker*: we only require that $Z$ is conditionally independent of a subset of features $X_P$ when conditioning on the remaining features $X_R$.
>
> We hope to better clarify the intuition behind CRDA and Assumption 1 below with an example...
> > Consider a dataset for predicting house prices.
> > The base model $g(x)$ captures systematic relationships (e.g., Square Footage -> Price).
> > The residual $z$ captures unobserved factors (e.g., a bidding war, the seller's urgency, or market sentiment).
> >
> > If we perturb a feature like *Garage Finish* (assuming it is identified as $X_P$), the systematic price $g(x)$ changes slightly.
> > However, a *bidding war* (the residual $z$) driven by buyers' urgency is likely independent of the garage finish.
> > Therefore, we can synthesize a valid new data point: a house with a different garage finish, adjusted systematic price, but the same *bidding war* noise driven by buyers' urgency.
> >
> > Conversely, if we perturbed *Neighborhood* ($X_R$), unobserved factors (like school district quality, which is part of $z$) would likely change, violating the invariance assumption.
> > **This is why our feature selection step is crucial.**
>
> We have added this example to Section 1 [Lines 40-45].
>
> ### 3 Iteratively with Retraining? (Q1)
>
> While iterative training is possible, we caution against it due to "bias amplification."
> Conceptually, we could iterate: train $g_1$ -> compute $z_1$ -> augment -> train $g_2$ -> compute $z_2$ -> augment... etc.
>
> There are a few issues here. First the augmented data is biased by the errors of $g_1$. If we train $g_2$ on that, and then generate new residuals from $g_2$, we risk refining the model on its own hallucinations (similar to "model collapse" in generative AI). Secondly, the residual $z$ is finite. Re-shuffling the same $z$ values onto different $X$ values iteratively doesn't add new information about the noise distribution, it only smooths the manifold of $g(x)$. Lastly, one could view this iterative process to be the same as just generating more samples from the first iteration.

---

> > ### Comment · Reviewer_S8VY · 2025-11-28
> >
> > Thank you for the explanation and discussion. My concerns have been fully addressed, and I feel positive about this submission. However, I noticed that I’m unable to update my score officially.

---

### Official Review · Reviewer_TQy8 · 2025-11-05

**Soundness:** 3
**Presentation:** 4
**Contribution:** 4
**Rating:** 6
**Confidence:** 3

**Summary:**

The paper proposes Counterfactual Residual Data Augmentation (CRDA), a novel, model-agnostic technique designed to enhance the robustness and generalization capabilities of models in tabular regression tasks characterized by limited training samples, high collection costs, and observational noise. The core principle relies on residual invariance: after training a base regressor to capture the behavior of the data, the remaining noise, or residual, is treated as an invariant component that remains stable under counterfactual perturbations of specific input features. New synthetic samples are generated by perturbing these selected features and assigning a counterfactual label preserving the original noise structure. CRDA includes checks for feature-residual independence and a validation safeguard via the Wilcoxon signed-rank test to ensure statistical significance before committing to the augmented dataset. Empirically, CRDA achieves substantial reductions in Mean Squared Error (MSE), averaging 22.9% for MLP regressors and 6.4% for XGBoost regressors, consistently outperforming state-of-the-art deep generative augmentation models.

**Strengths:**

The method successfully addresses its target domain, showing its greatest benefit in low to moderate data-scarce regimes (e.g., 2.5k to 20k samples in synthetic tests). This confirmation of a "sweet spot" is crucial, as is the observed average MSE reduction of 22.9% for the MLP Regressor, highlighting its ability to provide valuable signal for data-hungry neural models.

The inclusion of multiple checks acts as a filtering mechanism to prevent harmful augmentation.

The Wilcoxon signed-rank test on paired cross-validation errors provides a statistical safeguard where if no statistically significant improvement is found, CRDA reverts to the baseline model.
​
CRDA generates synthetic samples that are faithful to the underlying noise distribution, which the authors argue is the reason it reliably surpasses deep generative baselines.

**Weaknesses:**

While efficient in the "sweet spot," CRDA suffers at the boundaries of data scarcity. When the dataset is too small, the base model is too weak to produce meaningful, independent residuals, and the statistical tests may lack power. When the dataset is very large, the benefits diminish as the baseline is already highly accurate, limiting the practical application domain.

CRDA is designed for regression tasks limiting its immediate utility across general learning problems.

CRDA hinges on the Residual Invariance Principle (Assumption 1), which states that the noise must be conditionally independent of the perturbable features given the fixed features. In practice, this assumption is unverifiable from finite data. The paper acknowledges that Assumption 1 may be violated if the base predictor is poorly fitted (common in very small datasets) or if unobserved confounders are present, which are factors the initial PC algorithm check is ill-equipped to handle reliably.

**Questions:**

For very small datasets where the base model is weak and the Wilcoxon test lacks power, are there modifications to the methodology that could improve the fidelity of the residuals being generated, mitigating the risk of non-significant performance degradation?

have the authors explored conceptualizing the "residual" as uncertainty or confidence scores that could be sampled from an invariant distribution, rather than requiring a numerical difference from the true label?

How might CRDA's feature partitioning approach be integrated with causal inference techniques designed to handle proxies for unobserved confounders?

---

> ### Author Response · Authors · 2025-11-25
> **Rebuttal by Authors (Part 1)**
>
> ## Rebuttal (Part 1)
>
> We thank the reviewer for the careful reading and the constructive questions.
> Below we respond point-by-point (tags **W1–W3** for weaknesses, **Q1–Q3** for explicit questions).
>
> ### 1 Clarifying the Boundaries of Data Scarcity (W1)
>
> As acknowledged in Sec. 6, we agree with the reviewer that CRDA is most effective when the base model has learned a meaningful systematic component but is not yet saturated in accuracy. This is exactly the regime we target: low-to-moderate sample sizes where tabular regression is data-scarce but not "toy-sized."
>
> Empirically, our synthetic sample-size experiment (Fig. 3) already illustrates this behaviour. CRDA brings the largest MSE reductions between ≈2.5k and 20k samples, while offering less benefit at the two extreme ends of the sample sizes. The main experiments (shown in Table 1 / Fig. 2) which displayed consistent test MSE reductions, used 9 datasets of various sizes that are representative of many practical tabular regression tasks, thus we believe the practical domain of CRDA is still broad.
>
> Furthermore, we would like to clarify that although this is a limitation, we do not truly deem it a weakness, as it applies to every data augmentation technique. While some other works might not explicitly mention this, we are trying to be as transparent as possible. Every data augmentation mechanism *must* make some assumptions in order to leverage new information so that the new data will prove rich. Thus, all methods can suffer at the extreme ends of sample sizes where there is either not enough information or already sufficient enough information so improvement is not feasible.
>
> ### 2 Scope of CRDA (W2)
>
> We agree that CRDA, as presented, is designed for regression. Our goal in this work was to fully explore the residual-invariance idea in the simplest setting where the notion of a residual is canonical $(Y − g(X))$.
>
> Extending CRDA to classification is an exciting direction that we see as non-trivial rather than simple re-engineering. Hence, the current paper focused on regression to maintain a clear experimental scope and extensions to other learning problems are left as future work (mentioned in section 8 [Lines 520-523]).
>
> ### 3 Residual Invariance Assumption / Weak Model and Gates (W3 and Q1)
> We fully agree that Assumption 1 ("residual invariance") cannot be perfectly verified from finite data. This is inherent to any method relying on conditional independence assumptions. Our design goal was therefore not to *prove* Assumption 1 from data, but to combine (i) a cheap filter that rejects obviously problematic features and (ii) a data-driven gate that empirically tests whether augmentation actually helps.
>
> First, the PC-based graph check plus Pearson correlation filter are used only as a **risk-control heuristic** to identify candidate perturbable features. We emphasise in Section 6 that PC can indeed fail under unobserved confounding and non-linear dependencies; its main advantage is that, under standard assumptions, its edge error rate decays with sample size, which reduces the chance of accepting a truly bad feature as more data becomes available. If there are no candidate features identified then we revert to the baseline (Algorithm 1 lines 5–7).
>
> Second, and more importantly, CRDA includes a model-agnostic safeguard: if augmenting with the selected features does not yield a statistically significant improvement in cross-validated MSE (Wilcoxon signed-rank test, Algorithm 1 lines 15–19), we revert to the baseline.
>
> As a result of this design, CRDA already mitigates the risk of performance degradation when the base model is very weak or there is very little data. Even if the first screening (PC+Pearson) passes with candidate features and augmentation is attempted, the Wilcoxon signed-rank test will find no significant improvement (p ≥ α), so the baseline is again returned. The Wilcoxon gate actually *does not* lack power in smaller data settings since the p-value tends to become less precise and often *larger*, making it harder to pass our α threshold.
>
> Also, for weaker baselines, our additional study with linear regression baselines (Appendix G) illustrates how our safety nets help avoid degradation.
>
> Therefore, CRDA is deliberately conservative in extreme small-sample settings: it prefers abstention over applying potentially noisy residuals.

---

> > ### Author Response · Authors · 2025-11-25
> > **Rebuttal by Authors (Part 2)**
> >
> > ## Rebuttal (Part 2)
> >
> > ### 4 Residual as Uncertainty (Q2)
> >
> > The true residual indeed comes from an invariant distribution and our calculated numerical difference is merely a sample from it. While it is possible to explore the notion of uncertainty and confidence scores with CRDA, we feel that those are more readily applicable to classfiers where there is already a probability estimated for each class. When dealing with regression (the focus of our paper), where the targets are continuous, a numerical sample of the latent noise variable naturally fit our algorithm.
> >
> > ### 5 Proxies for Unobserved Confounders (Q3)
> >
> > Our work with CRDA actually already leverages a primitive form of this by using the residual $Z$ as a noisy proxy for unobserved confounders.
> >
> > Mathematically, if the true model is: $Y = f(X) + U + \epsilon$
> >
> > (where $U$ is the unobserved confounder and $\epsilon$ is random noise)
> >
> > Our residual is: $Z = Y - \hat{f}(X) \approx U + \epsilon$
> >
> > In scenarios where an unobserved confounder $U$ affects both $Y$ and $X$, the influence of $U$ on $Y$ is thus captured by the residual term $Z$. Consequently, our feature partitioning step (Algorithm 1, Step 3) exploits this: by testing for dependence between a candidate feature and $Z$, we are effectively testing for dependence between that feature and the latent confounder $U$. Features that correlate with this "proxy" ($Z$) are separated into $X_R$ to prevent invalid interventions.
> >
> > However, we do agree that integrating other methods from proximal causal inference is interesting future work that could prove beneficial and would amount to replacing our feature partitioning step. We have added a discussion on this in Section 8 [Lines 520-523].

---

### Note · Program_Chairs · 2026-01-17
**Submission Desk Rejected by Program Chairs**

The following references in this submission do not refer to real documents and/or have major errors in bibliographic information:

 Dhruv Mahajan, Saurabh Tripathi, Christopher Homan, Serge Belongie, and Pietro Perona. Counterfactual data augmentation for vision transformers. arXiv preprint arXiv:2303.17491, 2023.